# Overcoming efficiency and stability limits in water-processing nanoparticular organic photovoltaics by minimizing microstructure defects

Chen Xie[1], Thomas Heumüller[1], Wolfgang Gruber[2], Xiaofeng Tang[1], Andrej Classen [1], Isabel Schuldes[2], Matthew Bidwell[3], Andreas Späth[4], Rainer H. Fink [4], Tobias Unruh[2], Iain McCulloch[3,5], Ning Li [1] & Christoph J. Brabec[1,6]

There is a strong market driven need for processing organic photovoltaics from eco-friendly solvents. Water-dispersed organic semiconducting nanoparticles (NPs) satisfy these premises convincingly. However, the necessity of surfactants, which are inevitable for stabilizing NPs, is a major obstacle towards realizing competitive power conversion efficiencies for water-processed devices. Here, we report on a concept for minimizing the adverse impact of surfactants on solar cell performance. A poloxamer facilitates the purification of organic semiconducting NPs through stripping excess surfactants from aqueous dispersion. The use of surfactant-stripped NPs based on poly(3-hexylthiophene) / non-fullerene acceptor leads to a device efficiency and stability comparable to the one from devices processed by halogenated solvents. A record efficiency of 7.5% is achieved for NP devices based on a low-band gap polymer system. This elegant approach opens an avenue that future organic photovoltaics processing may be indeed based on non-toxic water-based nanoparticle inks.

[1] Institute of Materials for Electronics and Energy Technology (i-MEET), Department of Materials Science and Engineering, Friedrich-Alexander-Universität Erlangen-Nürnberg (FAU), Martensstrasse 7, 91058 Erlangen, Germany. [2] Institute for Crystallography and Structural Physics, Friedrich-Alexander-Universität Erlangen-Nürnberg (FAU), Staudtstrasse 3, 91058 Erlangen, Germany. [3] Department of Chemistry and Centre for Plastic Electronics, Imperial College London, London SW7 2AZ, UK. [4] Physical Chemistry 2 and ICMM Department of Chemistry and Pharmacy, Friedrich-Alexander-University Erlangen-Nürnberg (FAU), Egerlandstrsse 3, 91058 Erlangen, Germany. [5] King Abdullah University of Science and Technology (KAUST), KSC, Thuwal 23955-6900, Saudi Arabia. [6] Bavarian Center for Applied Energy Research (ZAE Bayern), Immerwahrstrasse 2, 91058 Erlangen, Germany. Correspondence and requests for materials should be addressed to C.X. (email: chen.xie@fau.de) or to N.L. (email: ning.li@fau.de) or to C.J.B. (email: christoph.brabec@fau.de)

The utilization of printing processes and non-toxic materials has made organic photovoltaics (OPVs) one of the most promising cost-effective and environmentally friendly solar cell technologies for the future energy supply[1–6]. However, the commonly used halogenated or aromatic solvents, such as chloroform (CF), chlorobenzene (CB), and 1,2-dichlorobenzene (DCB), stifles the large-scale production and widespread application of OPVs[7–12]. These toxic solvents unnecessarily increase the environmental impact of a future OPV technology[13]. Thus, today's fabrication of state-of-the-art OPV is not environmentally compatible, and significant effort is required to developing a well performing OPV concept, which can be processed from green solvents. OPV devices processed from non-halogenated & non-aromatic alternatives such as tetrahydrofuran (THF) and 2-methyltetrahydrofuran (MeTHF) have achieved power conversion efficiencies (PCEs) over 11%[14], and already indicate a great step into the right direction. However, THF is very limited in its ability to solubilize many of the current highest performing active materials, and the community is, therefore, missing a green concept that works for all material systems. Therefore, to push the OPV technology towards scalable processing for real-life applications, we have developed a concept for bulk-heterojunction (BHJ) nanoparticle (NP) inks with a frozen microstructure[15].

NP inks have been developed by dispersing organic materials into non-solvents through a miniemulsion technique[16,17]. Miniemulsion processing facilitates the fabrication of surfactant-stabilized aqueous BHJ dispersions consisting of conjugated polymer/fullerene composites. However, due to the residual stabilizers and improper distribution of the donor and acceptor domains[15,18–25], the performance of aqueous-processed NP solar cells was strongly limited to less than 4%, even for highly efficient donor:acceptor system[26]. Alternatively, combining another technique called nanoprecipitation[27] with green alcohols has successfully formed NPs without surfactants and achieved a PCE comparable to that of control devices based on conventional solution-processing. However, this concept only worked for BHJ systems based on poly(3-hexylthiophene) (P3HT) and fullerenes, where >4% PCE was reported for an indene-$C_{60}$ bisadduct (ICBA). Alternative polymeric absorbers other than P3HT did not form stable particle dispersions at sufficient high concentrations[28]. Therefore, a generic concept combining high-performance solar cell architectures with eco-friendly processing would be of tremendous value to the community.

Non-fullerene acceptors (NFAs) have recently boosted the efficiency of OPV devices to over 15%, due to their inherently better morphology in parallel to their widely tunable energy levels[29]. Owing to the excellent solubility in multiple solvent systems, we propose to investigate NFAs as a most promising alternative to fullerene acceptors for NP solar cells.

Here, we introduce an adapted nanoprecipitation technique[30] for NFA NP solar cells to overcome the PCE bottleneck of surfactant-terminated BHJ nanoparticles,. A micelle forming poloxamer (Pluronic F127) is selected to stabilize nanoparticles directly after precipitation. Most importantly, the excess surfactant density can be controlled by temperature modulation. Lowering the temperature allows to increase the critical micelle concentration (CMC) so that free and loosely bound poloxamers can be removed, leaving behind almost surfactant-free NPs for device processing. We deploy this processing approach to fabricate NP solar cells based on various polymer: NFA systems from water, and find that the otherwise surfactant limited charge transport, typically causing severe recombination in NP-based solar cells, can be overcome. NP solar cells based on P3HT and a rhodanine terminated small molecule acceptor o-IDTBR obtain a champion efficiency of 5.2%, while a record high PCE of 7.5% is achieved for the PBQ-QF:ITIC NPs, which is much higher than the values reported previously (Fig. 1a, Supplementary Fig. 1. and Supplementary Table 1). The elegant surfactant-stripping approach reported in this work introduces a generic concept for

aqueous-processed NFA-based OPV devices with promisingly high-efficiency and stability.

## Results

**Synthesis of NFA-based surfactant-stripped nanoparticles.** In our previous work, solar cells based on a NFA, namely IDTBR, could deliver state-of-the-art efficiencies and long-term stability in various donor systems[31,32]. However, the microstructure morphology of IDTBR-based OPV devices is crucial for reaching high-performance, and meticulous device optimization is therefore required. Moreover, the commonly used organic solvents and solvent additives, such as CB, DCB and diiodooctane (DIO), are hazardous and cannot be implemented for large-scale industrial production. To further push the promising NFA-based OPV technology towards commercialization, we employed IDTBR for fabrication of water-processed NP solar cells. However, when replacing ICBA with IDTBR, we found that nanoprecipitation (Supplementary Note 1) fails to give high qualitative particles as required for solar cell fabrication. As shown in Supplementary Table 2 and Supplementary Fig. 2, P3HT:IDTBR was capable of forming NPs in some "green" alcohols, but rather large aggregations were observed from scanning electron microscope (SEM) imaging and the large phase separation revealed by incomplete photoluminescence (PL) quenching in NP-processed films, indicating that the NFA-based NPs cannot be directly used for the fabrication of efficient organic solar cells. Thus, a stabilizer is necessary to control the aggregations and to preserve the required microstructural morphology. Without stabilizers, it has not been able to stabilize nanoparticles in sufficiently concentrated solutions, disqualifying them for the fabrication of NP OPVs[33].

It is, therefore, inevitable to find a proper stabilizing strategy that can form NFA nanoparticles for solar cell fabrication. Some popular stabilizers being used for conjugated polymer nanoparticle synthesis in diverse photoelectronics, including sodium dodecyl sulfate (SDS), dodecyltrimethylammonium bromide (DTAB), polyethylene glycol dodecyl ether (Brij), poly(styrene-alt-maleic acid) (PSMA), poly(methacrylic acid) (PMAA), and Pluronic F127, were employed to prepare NFA-based NPs in water (Supplementary Fig. 3)[34–38]. We found that the nanoprecipitation approach based on the poloxamer Pluronic F127 (Fig. 2a), otherwise used as a surfactant for biological imaging[39–41], is indeed compatible to the synthesis of stable NPs dispersions. The synthesis route for NPs in the presence of F127 is described as follows: the organic composites, donor polymer and NFA, were first dissolved in THF. This solution was rapidly added into an aqueous F127 solution under sonication. The sudden change of solubility caused precipitation of the hydrophobic conjugated semiconductors and finally formed a NP dispersion[30,42]. In order to form well-distributed NPs, good solubility in THF is the first consideration for the active materials. Due to the inferior solubility of fullerene in THF, small molecule NFAs such as o-IDTBR (Fig. 2a) are excellent candidates to replace fullerene-based acceptors for this technique.

The poloxamer F127 was selected because of its temperature-sensitive CMC[43,44]. As described in Fig. 2b, when we lowered the temperature of the NP dispersion to nearly 0 °C, a steep increase of CMC converted F127 micelles back to linear poloxamers in water and enabled removal of all excess stabilizers by centrifugal filtration (described in Methods), leaving retentate with rather pure BHJ NPs behind. The filtrate after each centrifugal wash was collected for evaluating the quantity of residual F127 during such surfactant-stripping process (described in Methods)[39]. As shown in Fig. 2c, a fivefold centrifugal segregation removed all excess F127 at 0 °C, as determined by colorimetric assay (Supplementary Fig. 4)[45], whereas the process is ineffective at 25 °C. The first wash only removed 8% of F127 when it was done at room temperature, which is much less than over 60% of elimination at 0 °C. This

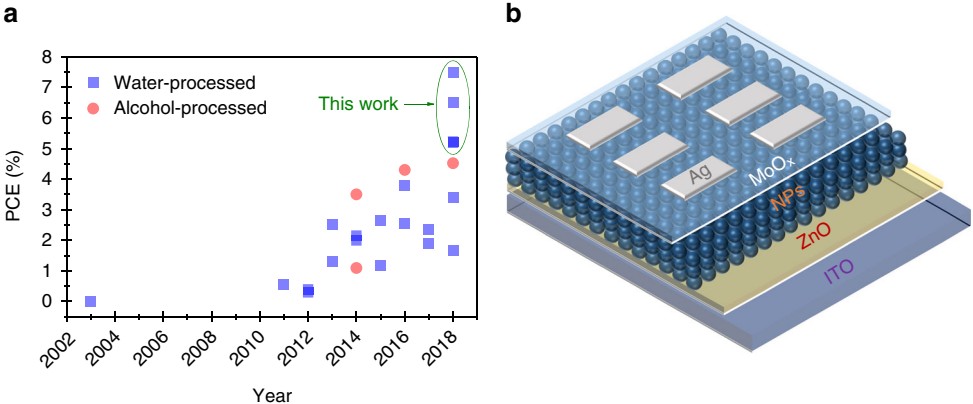

**Fig. 1** Water/alcohol-processed NP solar cells. **a** Efficiency evolution of NP-based organic solar cells processed by water/alcohol. **b** Device architecture of NP-based organic solar cells in this work

surfactant-stripping technique did not lead to a sedimentation of active materials, which were 100% retained during the washing process. Unlike the P3HT:o-IDTBR NPs in alcohols, no large aggregates are observed from the surfactant-stripped NPs with a controlled surfactant density (cs-NPs) by SEM (Fig. 2d). The constancy of averaged size of cs-NPs (Fig. 2e) demonstrates that the cs-NPs are highly stable and no occurrence of Ostwald ripening[46] and only minor sedimentation was observed after three months storage at around 0 °C.

The uniqueness of this strategy was further proven by measuring the surface charge as a function of washing steps. As shown in Fig. 2f, centrifuging the NP dispersions with all types of surfactants at room temperature took 5 to 6 times washing before the surface potential began to saturate, including F127. Nevertheless, the surfactant-stripping process at 0 °C decreased the surface charge after 2 to 3 times washing to a saturation value of about −10 mV (Supplementary Table 3), which is a significantly lower value. As the reference surfactants are not amenable to low-temperature surfactant-stripping, similar zeta potential values were recorded for room temperature as well as 0 °C. Colorimetry assay[40] evidenced that the quantity of tightly bound F127 left on the particle surface is reduced to about 2 % of the original amount after 0 °C washing (Fig. 2g and Methods). In direct comparison, the retention of SDS was found to be more than 13% even after ten times washing (Supplementary Fig. 5 and Note 2). Therefore, the cs-NP dispersion based on this CMC-switching strategy is expected to give films with highest possible purity, which we already have evidenced to be beneficial for the performance of optoelectronic devices[36].

**Water-processing of cs-NP film.** Having established F127 as the most promising surfactant forming a stabilizer-stripped NP dispersion, we further examined the film formation properties of BHJ cs-NPs. In a recent industrial Figure of Merit (i-FoM) analysis, we have reported that P3HT would be the most promising donor for commercialization of NFA-based solar cells, owing to its promising high stability and extremely low synthetic complexity[47]. Hence, the P3HT:o-IDTBR films were first investigated and optimized using F127-stabilized NPs. Near-edge x-ray absorption fine structure (NEXAFS) spectroscopy demonstrates that the ratio between P3HT and o-IDTBR does not change during the NP synthesis (Supplementary Fig. 6 and Note 3). Typically, the water processing of NP does not improve the film formation. Thus, thermal annealing was applied to overcome the inhomogeneity of as-cast NP layers (Supplementary Fig. 7). Supplementary Fig. 8 exhibits that the amount of residual F127 significantly affects the microstructure formation of the deposited thin films. As shown in Fig. 2g, the incomplete PL quenching in annealed films is a signal of obstructed charge transfer between polymer and NFA by residual F127. These results suggest that the surfactant-stripping technique could minimize the impact of residual surfactant on film morphology during the film formation.

Moreover, we fabricated three types of P3HT:o-IDTBR thin films under identical processing conditions to study the difference in microstructural morphology: spin-coated BHJ films processed from THF solution (30 mg mL$^{-1}$), which is the precursor solution for cs-NP synthesis, were compared to SDS-stabilized NPs and F127-stabilized cs-NP films. Grazing-incidence wide-angle X-ray scattering (GIWAXS) measurements were carried out to study the formation of water-processed NP films. Figure 3 shows the GIWAXS profiles of thin films before and after thermal annealing collected from out-of-plane and in-plane cuts. cs-NP films formed a significantly more ordered P3HT phase as compared to SDS-stabilized NP films, which is evidenced by strong (100) and (200) out-of-plane diffraction peaks in the two-dimensional (2D) image (Supplementary Fig. 9)[48]. Meanwhile, the relatively weakened diffraction peaks from out-of-plane cuts has also been observed in F127-stabilizd NP film with incomplete washing process (Supplementary Fig. 10). NP dispersions do not show preferential orientation or anisotropic differences in crystallinity as evidenced by transmission wide-angle X-ray scattering (WAXS) and transmission small-angle X-ray scattering (SAXS) (Supplementary Note 4). However, AFM measurements evidence deformation of nanoparticles during film formation (Supplementary Fig. 12 and Note 5), suggesting geometrical rearrangement of the single particles during spin-coating or drying. After annealing at 150° C, additional crystalline features emerged in the diffractogram (Supplementary Fig. 9d–f) and the corresponding peaks were observed at in-plane cuts for all three films (Fig. 3d). Those peaks centered at $Q_y = 4.27$ nm$^{-1}$, 5.90 nm$^{-1}$, and 11.9 nm$^{-1}$ match well with those from o-IDTBR (Supplementary Fig. 13). These peaks most probably result from face-on crystallization of NFA aggregates during thermal treatment. In addition, the orientation of P3HT has turned from preferential edge-on orientation to a more isotropic one. Annealing increases face-on crystallinity in all three films but otherwise results in similar crystallization trends.

Summarizing the above observations, we suggest the following model to rationalize the shape of NPs after film deposition in Supplementary Fig. 14. NP touching the substrate surface during spin-coating become deformed to considerable shear forces, resulting from the difference in adhesive and centrifugal forces.

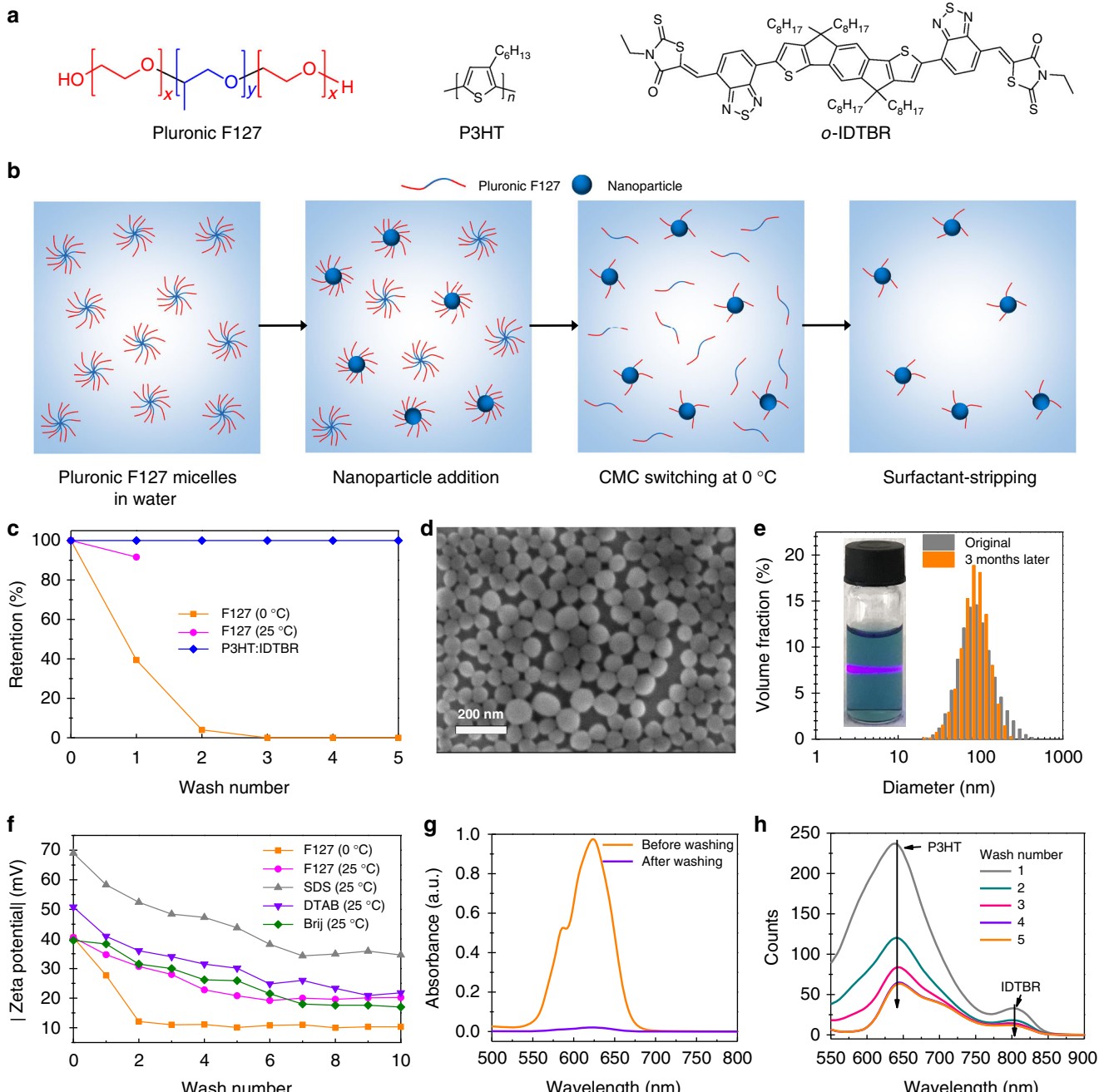

**Fig. 2** Polymer:non-fullerene acceptor cs-NPs generated from surfactant-stripping technique. **a** Chemical structure of polymer (P3HT), NFA (o-IDTBR), and surfactant (Pluronic F127) for cs-NP synthesis. **b** Synthesis and purification of nanoparticles by temperature-mediated CMC-switching strategy. **c** F127 retention as function of centrifugal washes at freezing and room temperature in water. **d** SEM of dried P3HT:IDTBR cs-NPs. Scale bar, 100 nm. Sample was prepared by spin-coating 2.5 mg mL$^{-1}$ of aqueous cs-NP dispersion on silicon slide. **e** Tyndall effect observed from P3HT;IDTBR cs-NPs dispersion and size distribution of NPs before and after storage by DLS. The dispersion sample was kept in a freezer at approximately 0 °C for 3 months. **f** Zeta potential of NP dispersions synthesized by various surfactants during centrifugal washes. **g** Retention of tightly bounded F127 manifested by the absorption of ammonium thiocyanate salt-F127 complex (see Methods). **h** PL spectra (Excited at 500 nm) of annealed (150 °C, 10 min) films (approximately 200 nm) deposited by cs-NPs with different centrifugal washes at freezing temperature

Residual surfactants on the NP surface hamper solid state recrystallization between single nanoparticles. On the other hand, surfactant-free NPs can undergo reorganization and eventually recrystallization at their grain boundaries. Overall, we highlight that cs-NP films suffer significantly less disturbance from the surfactant and exhibit significantly higher crystallinity than films from SDS-NP.

**Characterization of cs-NP-based organic solar cells**. We fabricated NP-based solar cells by aqueous dispersions in the inverted device configuration (Fig. 1b). The surfactant-stripping technique is superior for fabrication of NFA-based NP solar cells, and the corresponding photovoltaic parameters are summarized in Table 1. The optimized cs-NPs-processed P3HT:o-IDTBR devices exhibit a $V_{OC}$ of 0.76 V, a $J_{SC}$ of 10.7 mA cm$^{-2}$ and a high FF of

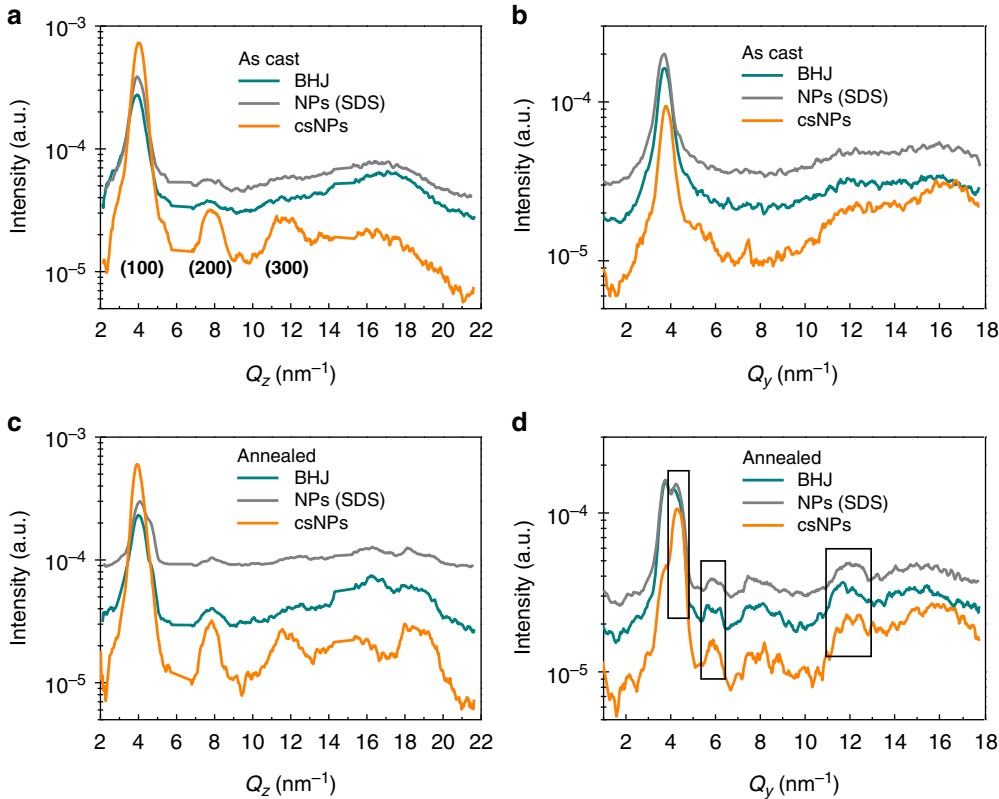

**Fig. 3** Grazing-incidence wide-angle X-ray scattering (GIWAXS) of cs-NP film. The GIWAXS profiles of P3HT:o-IDTBR obtained from BHJ film processed from THF solution, NP (SDS)-based film and cs-NP-based film, respectively. As cast films collected from **a** out-of-plane cuts at $Q_y = 0$ and **b** in-plane cuts at $Q_z = 0$ as well as annealed films (150 °C, 10 min) collected from **c** out-of-plane cuts at $Q_y = 0$ and **d** in-plane cuts at $Q_z = 0$

| Table 1 Photovoltaic parameters of solar cells processed by solutions or NP dispersions | | | | | | |
|---|---|---|---|---|---|---|
| **Material** | **Solvent (surfactant)** | **Thickness [nm]** | **$V_{OC}$ [V]** | **$J_{SC}$ [mA cm$^{-2}$]** | **FF [%]** | **PCE [%]** |
| P3HT:o-IDTBR | THF | 210 | 0.74 ± 0.02 | 8.38 ± 0.40 | 48.9 ± 2.5 | 3.11 ± 0.34 (3.45) |
| | Water (SDS) | 210 | 0.75 ± 0.01 | 7.45 ± 0.31 | 45.2 ± 2.1 | 2.53 ± 0.20 (2.73) |
| | Water (F127) | 230 | 0.76 ± 0.01 | 10.36 ± 0.39 | 62.9 ± 2.0 | 4.95 ± 0.32 (5.23) |
| PCE10:o-IDTBR | Water (SDS) | 90 | 0.97 ± 0.03 | 9.20 ± 0.32 | 40.7 ± 2.9 | 3.61 ± 0.48 (4.12) |
| | Water (F127) | 100 | 0.97 ± 0.03 | 12.01 ± 0.43 | 42.4 ± 2.0 | 4.94 ± 0.25 (5.19) |
| PBQ-QF:o-IDTBR | Water (SDS) | 90 | 0.92 ± 0.04 | 9.79 ± 0.80 | 38.3 ± 2.0 | 3.45 ± 0.57 (4.02) |
| | Water (F127) | 90 | 0.95 ± 0.03 | 13.09 ± 0.41 | 47.9 ± 4.3 | 5.96 ± 0.58 (6.52) |
| PBQ-QF:ITIC | Water (SDS) | 110 | 0.84 ± 0.01 | 10.31± 0.32 | 46.0 ± 3.1 | 3.98 ± 0.49 (4.42) |
| | Water (F127) | 110 | 0.85 ± 0.02 | 14.87 ± 0.30 | 52.7 ± 2.9 | 6.97 ± 0.53 (7.50) |

0.65, resulting in a champion PCE of 5.23 % (Fig. 4a). It is notable that the cs-NP-based devices delivered superior photovoltaic performance to the BHJ devices processed from THF solution, and are comparable with the performance of optimized devices processed from halogenated solvents and solvent mixtures (Supplementary Table 1)[31,49]. Furthermore, the PCE of cs-NP device was found to be thickness independent within an active layer between 60 and 300 nm (Supplementary Fig. 15). The centrifugal washing (described in Methods) played a crucial role in achieving high-performance for devices with F127-based NPs. As shown in Supplementary Fig. 16, the losses in FFs and PCEs are observed from the solar cells processed by water dispersions with residual F127[15,24].

To study the origin of the very promising $J_{SC}$ and FF values found for cs-NP devices as compared to the conventional NP devices, the charge generation behavior was investigated firstly. The photocurrent ($J_{Ph}$) vs. field scaling in Fig. 4c indicates that $J_{Ph}$

quickly saturates under low effective reverse bias ($V_0 - V$) in cs-NP-based device[50]. Nonetheless, the THF and water (SDS)-processed devices exhibit a stronger field-dependence of $J_{Ph}$. The internal quantum efficiency (IQE) measured under different bias (Fig. 4d, with the single spectra in Supplementary Fig. 17) confirm that cs-NP-based devices show a lower bias dependence of charge extraction. The maximum charge carrier generation rate $G_{MAX}$ (Table 2) of those three systems indicates that the charge generation does not dominate the losses in those two control devices. The saturation current density is defined by $J_{sat} = qG_{MAX}L$, where $q$ is the elementary charge and $L$ is the active layer thickness (Table 1).

Another method was introduced to understand the loss mechanism in THF and water (SDS) processed devices. The exciton harvesting efficiency ($\eta_{eh}$) and charge collection efficiency ($\eta_{cc}$) were calculated based on the results of PL and IQE results

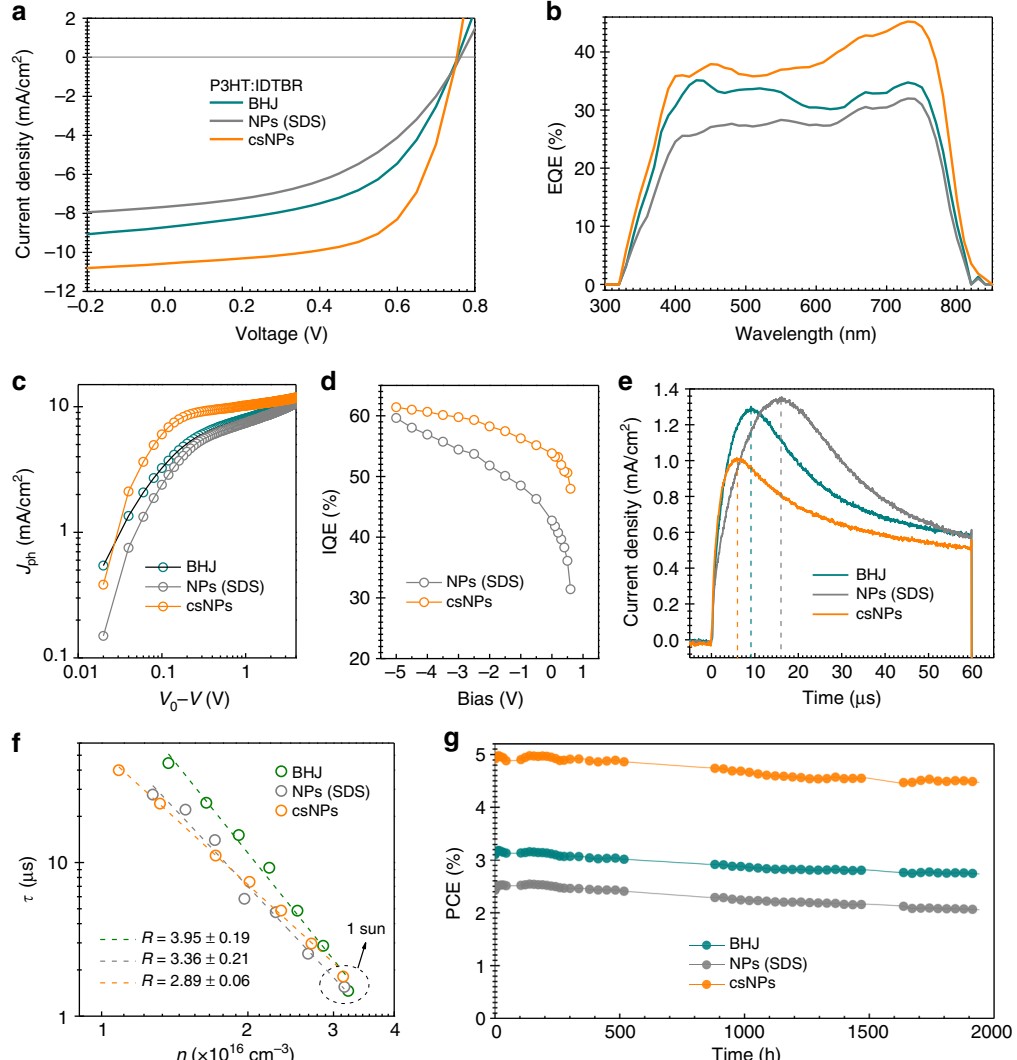

**Fig. 4** Device characteristics and stability of cs-NP-based organic solar cells. **a** Light current–voltage, **b** EQE, **c** photocurrent as function of effective voltage, **d** IQE values at wavelength of 730 nm as function of applied bias, **e** Photo-CELIV traces after a delay time of 1 μs, **f** Charge carrier lifetime $\tau$, obtained from TPV, as function of charge density $n$, calculated from CE measurements under $V_{OC}$ conditions and **g** PCE in the course of 2000 h light exposure with UV filter in $N_2$ atmosphere of P3HT:IDTBR-based solar cells processed by BHJ THF solution, water dispersion with NPs (SDS) and water dispersion with cs-NPs. Device architecture: glass/ITO/ZnO/active layer/MoO$_x$/Ag

**Table 2 Parameters from device characterization**

|  | $J_{sat}$ [mA cm$^{-2}$] | $G_{MAX}$ [cm$^{-3}$ s$^{-1}$] | $\mu$ [cm$^2$ V$^{-1}$ s$^{-1}$] | $\beta$ [cm$^3$ s$^{-1}$] | $\tau$ [μs] | $\lambda$ |
|---|---|---|---|---|---|---|
| THF | 11.97 | $3.44 \times 10^{-21}$ | $1.14 \times 10^{-4}$ | $2.61 \times 10^{-13}$ | 1.46 | 2.95 |
| Water (SDS) | 11.25 | $3.35 \times 10^{-21}$ | $6.88 \times 10^{-5}$ | $1.89 \times 10^{-13}$ | 1.45 | 2.36 |
| Water (F127) | 12.01 | $3.26 \times 10^{-21}$ | $4.04 \times 10^{-4}$ | $1.72 \times 10^{-13}$ | 1.81 | 1.89 |

Summary of calculated charge generation, collection, transport and recombination parameters in optimized P3HT:o-IDTBR solar cells: saturation current ($J_{sat}$); maximum rate of free charge carrier generation in saturation regime ($G_{MAX}$); charge carrier mobility ($\mu$) and bimolecular recombination coefficient ($\beta$) from photo-CELIV measurements; charge carrier lifetime ($\tau$), and recombination exponent ($\lambda$) from TPV&CE measurements

(Supplementary Note 6). The cs-NP device exhibits an estimation of both slightly higher $\eta_{eh}$ and $\eta_{cc}$ as compared to both control devices, indicating a relatively efficient exciton splitting and charge collection in cs-NP system.

We further investigated the charge carrier mobility to see whether the surfactant would affect the transport. The photo-induced charge carrier extraction by linearly increasing voltage (Photo-CELIV)[51]

measurement in Fig. 4f shows that the cs-NP device achieved an averaged mobility of $4.04 \times 10^{-4}$ cm$^2$ V$^{-1}$ s$^{-1}$, being 4–6 times higher than that from BHJ ($1.14 \times 10^{-4}$ cm$^2$ V$^{-1}$ s$^{-1}$) and NP (SDS) ($6.88 \times 10^{-5}$ cm$^2$ V$^{-1}$ s$^{-1}$) devices. Further measuring photo-CELIV under various delay time ($t_d$) (Supplementary Fig. 21) and the calculation of bimolecular recombination coefficient ($\beta$) provided insight into the recombination kinetics in NP solar cells

(described in Methods)[52]. As listed in Table 2, the cs-NP device has the lowest calculated $\beta$ among all those systems, confirming the reduced second order recombination combined with increased charge carrier mobility[53].

For investigation of detailed recombination mechanism, the charge carrier lifetime ($\tau$) and charge carrier density ($n$) were studied via transient photovoltage (TPV) combined with charge extraction (CE) techniques (described in Methods)[54]. Fig. 4g compares the $\tau$ as a function of the $n$ for the NP systems. Although the three systems exhibit similar values of $n$ and quite comparable carrier lifetimes at 1 sun (Table 2), the recombination order $R = \lambda + 1$ shows different behaviors. For these two control systems, the charge carrier decay dynamics exhibits the relatively high order of charge density dependency ($R = 3.95 \pm 0.19$ for THF-processed BHJ device and $R = 3.36 \pm 0.21$ for NP (SDS) device) compared to the cs-NP device ($R = 2.89 \pm 0.09$). A recombination order R higher than two is attributed to the trapping and release effects in energetic traps and morphological traps as in the case of an inhomogeneous donor-acceptor distribution[55]. Considering the residual SDS, the higher trap-effect in NP (SDS) device is plausible. The high order of recombination with increased energetic disorder can deteriorate the FF and result in a suppression of injection (or increased $R_S$, Supplementary Table 5)[56]. The THF-based BHJ device suffers from trap-effect, but its higher mobility than that of NP (SDS) ones suppresses the losses in $J_{SC}$ and FF. It is worthwhile to note that we previously reported an $R = 2.1$ for highly optimized P3HT:o-IDTBR with over 6% PCE when spin-coated from halogenated solvents. Light intensity-dependent $V_{OC}$ measurements of cs-NP device suggest slightly reduced first-order recombination compared to the control devices[57–59] (Supplementary Fig. 22) We believe that further exploring the parameter space of the surfactant-stripping process may allow us to better understand controlling the donor/acceptor microstructure on the level of single nanoparticles.

To summarize, the significant enhancement of FF and $J_{SC}$ in cs-NP devices is mainly attributed to the increased charge carrier mobility and the decreased energetic disorder by overcoming the influence of residual stabilizers and incomplete crystalline, which lead to water-processed devices with photovoltaic performances comparable to those processed from halogenated solvents and solvent mixtures[31,49].

We further investigated the stability of NP-based devices under illumination provided by a metal halide lamp (a ultraviolet (UV) cut-filter at 380 nm is used) under nitrogen atmosphere[60]. As shown in Fig.4i and Supplementary Fig. 23, the cs-NP-based device with a PCE around 5% exhibited a performance loss of only 8% after 2000 h lifetime testing under 1 sun irradiation. Similar to the P3HT:IDTBR-based devices processed from halogenated solvents[60], no photoinduced "burn-in" loss[61] was observed in water-processed solar cells, indicating the absence of film disorder under illumination. These results suggest that the water-processing does not negatively influence the intrinsic stability of P3HT:o-IDTBR solar cells.

**University of surfactant-stripping technique**. To prove the generality of the approach, a low-bandgap polymer PBQ-QF was further investigated due to its good solubility in THF[62]. Two NFAs, o-IDTBR and ITIC, were combined with this polymer for cs-NP synthesis (Fig. 5a). As shown in Supplementary Fig. 24, both PBQ-QF:o-IDTBR and PBQ-QF:ITIC cs-NPs were well controlled with a size distribution ranging from 60 to 80 nm. As expected, the surfactant-stripping approach was found to result in highly efficient organic solar cells based on these two NP systems. The water-processed PBQ-QF:o-IDTBR cs-NP devices without optimization achieved a PCE of

6.5% (Fig. 5c). A more promising PCE of 7.5% along with a $V_{OC}$ of 0.87 V, a $J_{SC}$ of 15.4 mA cm$^{-2}$ and a FF of 0.55, was obtained for PBQ-QF:ITIC-based cs-NP devices (Fig. 5d). Although the FFs are a little inferior than those of the optimized P3HT system, further morphology modifications on NP size, film thickness and annealing temperature would overcome the charge collection problems in those low-bandgap systems. The high Jsc value was confirmed by external quantum efficiency (EQE) integration (Supplementary Table 5), which showed a very impressive maximum value of 74% at 650 nm (Fig. 5g). In addition, cs-NP based on another THF-soluble polymer PBDTTT-EFT (PTB7-Th or PCE10) exhibited a promising PCE of 5.19% in combination with o-IDTBR (Fig. 5a). In line with the findings for P3HT, corresponding NPs (SDS) processed devices consistently showed a drop in $J_{SC}$ and FF. The azimuthal integration of (100) GIWAXS diffraction peak demonstrates that all those 4 polymer:NFA systems exhibit higher crystallinity for cs-NP films than for SDS-NP ones (Supplementary Fig. 27). Based on the device characteristics and morphology investigation, the probable morphology and charge transport in cs-NP films can be schematically illustrated in Fig. 5h. As observed from the P3HT:o-IDTBR cs-NP system, the surfactant-stripping removes most of the surfactants and results in more ordered polymer and NFA domains. In contrast, the residual surfactant in non-annealed SDS-NP films deteriorates crystallinity (Fig. 3a) and most likely is responsible for the reduced mobility (Fig. 4f) along with the higher trap-effect (Fig. 4g). These results exhibit the potential of eco-friendly water-processing approach to catch up with the photovoltaic performance of state-of-the-art BHJ OPV devices.

The stability data of the three systems under continuous one sun illumination is shown in Supplementary Fig. 28. Both IDTBR-based solar cells presented PCE losses <5% after 1000 h illumination, while a "burn-in" degradation (approximately 15%) was observed for PBQ-QF:ITIC-based cs-NP solar cells.

An impressive device performance and stability was achieved via surfactant-stripping for these systems. However, we noticed that this technique might be incompatible with some material systems, for instance materials with very low solubility in THF and materials suffered from strong aggregations during NP synthesis. Therefore, the choice of materials with high solubility in THF is one essential consideration for a successful cs-NP synthesis and device fabrication.

**Discussion**

In summary, we successfully demonstrated a unique and elegant approach to synthesizing water-dispersed NPs for fabrication of high-performance OPVs. A concept based on a micelle forming poloxamer was demonstrated to stabilize NPs that could be subjected to low-temperature CMC-switching and surfactant stripping. This approach enables the synthesis of highest purity light-harvesting NPs by minimizing the amount of residual surfactant in aqueous system. This surfactant-stripping technique requires no excessive chemical synthesis for water-soluble polymers and is readily applicable to various NFA-based systems. Due to a higher crystallinity and a reduced defect density from surfactants, polymer:NFA-based solar cells processed from the surfactant-stripped cs-NPs have overcome the charge transport and recombination limits typically observed for the NP devices with traditional surfactant (SDS). A champion PCE of 5.23% with a high FF of 0.65 was achieved for P3HT:o-IDTBR NP solar cells, being comparable to the devices processed from the halogenated solvents. Moreover, cs-NPs based on the low-bandgap polymer PBQ-QF further boosted the record PCE of water/alcohol-processed OPV up to 7.5%. The demonstration of cs-NP synthesis offers a smart strategy towards industrial mass production of OPV devices with high-efficiency and stability from eco-friendly aqueous solvent system. We believe that, this technique can be

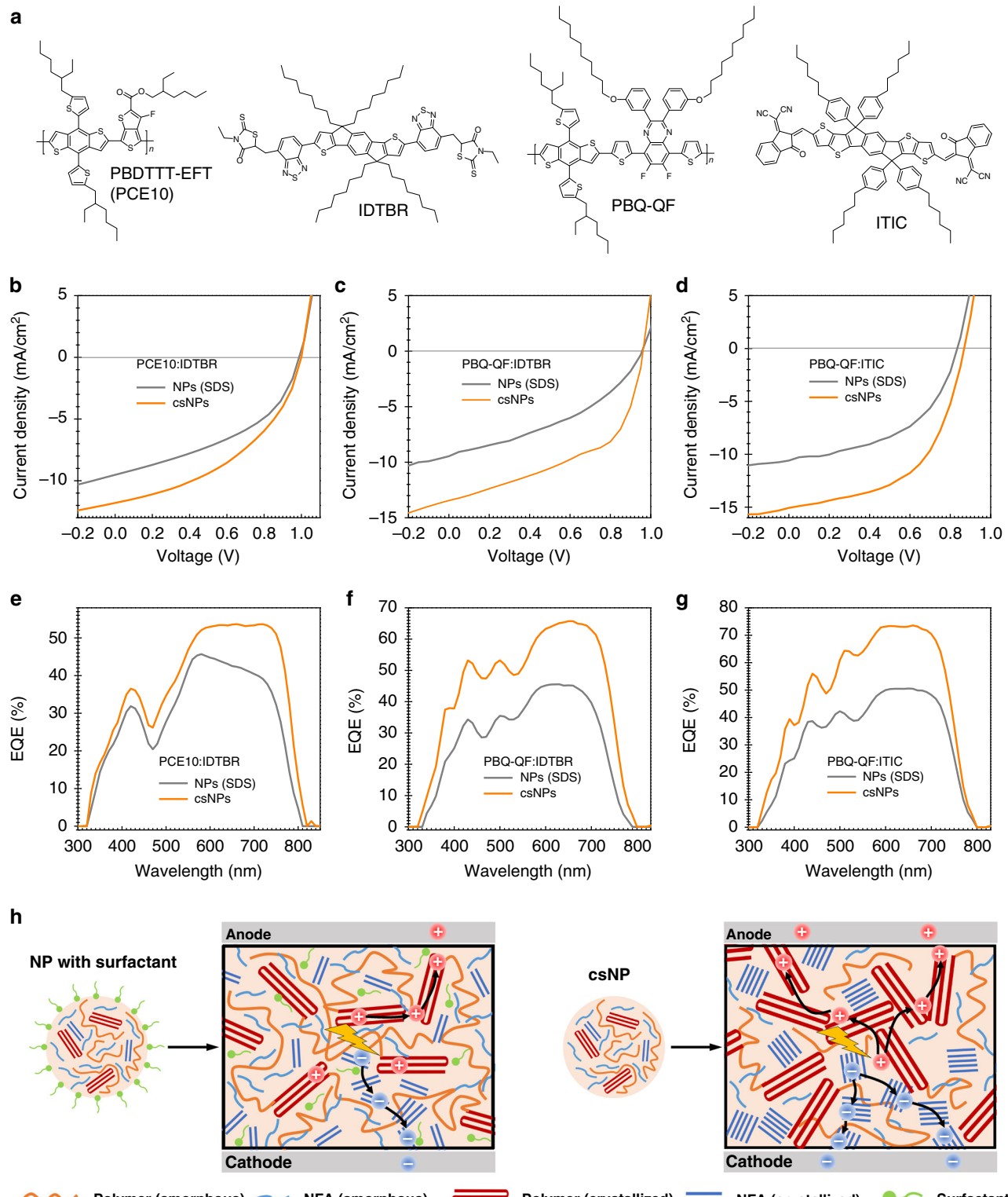

**Fig. 5** Generality of surfactant-stripping technique for polymer:NFA solar cells. **a** Chemical structure of donors and acceptors for generality test of surfactant-stripping technique. **b**, **c**, **d** Light current–voltage characteristics and **e**, **f**, **g** EQE of **b**, **e** PCE10:IDTBR, **c**, **f** PBQ-QF:IDTBR and **d**, **g** PBQ-QF:ITIC-based solar cells processed by water dispersion with NPs (SDS) and cs-NPs; **h** Schematic diagram of NP (SDS) and cs-NP structures as well as morphology and charge transport in their corresponding NP devices

transferred to the large-scale fabrication through the combination of mass synthesis, cross-flow ultrafiltration[63] and roll-to-roll printing. In addition, by means of low-cost P3HT:NFA system[64] with cs-NP processing, it is expected to achieve commercially viable OPVs with the cost around 10 € g[−1][65]. Further, we are

confident that this generic concept of cs-NP is potentially viable for processing other water-based organic photoelectronics with high performances, such as organic field effect transistors (OFETs), organic light-emitting diode (OLED) and organic photodetectors.

## Methods

**Materials.** P3HT ($M_w = 70$ K, RR = 96), PBQ-QF and o-IDTBR were purchased from Merck, Advent Materials and Flexink, respectively. PCE10 and ITIC were purchased from 1-Material, Pluronic F127, sodium dodecyl sulfate (SDS), Dodecyltrimethylammonium bromide (DTAB), poly(styrene-co-maleic anhydride) (PSMA) and all the solvents were from Sigma Aldrich, poly(methacrylic acid) (PMAA) from American Dye Source (Quebec, Canada). All the materials were used as received without further purification. Zinc oxide (ZnO) nanoparticles were purchased from Avantama.

**NP synthesis.** Mini-emulsion synthesis of polymer/non-fullerene acceptor NPs: Thirty milligram donor/acceptor mixture was dissolved in 1 mL chloroform and added to 6 mL 10 mg/mL SDS or DTAB aqueous solution and stirred for 2 h. The formed micro-emulsion dispersion was ultrasonicated using a Hielscher UPS200S ultrasonic finger in a water bath. The miniemulsion system was heated at 70 °C while constant stirring until chloroform was completely eliminated. The excess surfactant from the particle solution was removed using Amicon® ultra-15 centrifuge filter (cutoff 30K). The dispersion was placed into the filter and centrifuged at 4000 rpm for 20 min The retentate was raised to 15 mL with water and then centrifuged again. This process was repeated for several times until surface tension of the filtrate reached $38 \pm 2$ mN m$^{-1}$. The retentate was filtered by a 0.45 µm filter before the last centrifugation.

Nanoprecipitation synthesis of polymer/non-fullerene acceptor NPs: For the particle with surfactant F127 or Brij, 5 mg donor/acceptor mixture was dissolved in 1 mL tetrahydrofuran and injected into 2 mL 20 mg mL$^{-1}$ F127 or Brij aqueous solution in a sonic bath. A stream of N$_2$ gas went through the solution to remove THF.

For the particle with PSMA, 100 µg donor/acceptor mixture and 20 µg PSMA was dissolved in 1 mL THF and injected into 2 mL water in a sonic bath. A stream of N$_2$ gas went through the solution to remove THF.

For the particle with PMAA, 100 µg donor/acceptor mixture was dissolved 50 µL THF and injected into 2 mL (5 mg mL$^{-1}$) PMAA aqueous solution in a sonic bath. This precipitation process was repeated multiple times to increase the nanoparticle concentration. A stream of N$_2$ gas went through the solution to remove THF.

Surfactant stripping after nanoprecipitation synthesis: After THF evaporation, the NPs stabilized by F127 were cooled to 0 °C. Then the dispersion was placed into Amicon® ultra-15 centrifuge filter (cutoff 100K) and centrifuged at 4000 rpm for 15 min The retentate was raised to 15 mL with water and then centrifuged again. This process was repeated for several times. The retentate was filtered by a 0.45 µm filter before the last centrifugation. This protocol does not work for other surfactants such as PSMA and PMAA. PSMA is insoluble in water and PMAA has too large molecular weight.

**F127 quantification.** A reported colorimetry assay[45] based on the capability of cobalt thiocyanate to form a complex with poloxamer polymers was used to determine the concentration of F127. In our protocol, 0.3 g cobalt nitrate hexahydrate and 1.2 g ammonium thiocyanate was dissolved in 3 mL water for cobalt thiocyanate preparation. Three-hundred microliters solution of cobalt thiocyanate was then combined with 120 µL F127 solution with the concentration range from 1 to 20 mg mL$^{-1}$, 600 µL ethyl acetate and 240 µL ethanol. After gently stirring, the mixture was centrifuged at 12,000 rpm for 5 min. Then the blue supernatant was removed using pipette and leaving blue pellet in the tube. The blue pellet was washed using ethyl acetate several times until the supernatant became colorless. The pellet was then redissolved in 1 mL of acetone to measure the absorbance at 623 nm. Supplementary Fig. 4 shows the calibration curve of the absorbance of cobalt thiocyanate-F127 complex as function of the concentration of F127. The F127 retention was obtained by collecting the filtrate after every washing step and calculating the concentration with this curve. The retention of P3HT:o-IDTBR was determined by measuring absorbance.

For the quantification of tightly bound F127 in cs-NPs (Fig. 2g), 120 µL dispersion with 20 mg mL$^{-1}$ F127 was mixed with 240 µL ethanol and stirred until a complete precipitation of all light-harvesting materials. The sediment was then removed by a filtration through a 0.45 µm filter and obtaining a colorless filtrate. Then 300 µL aqueous cobalt thiocyanate solution combined with 600 µL ethyl acetate were added. After gently stirring, the mixture was centrifuged at 12,000 rpm for 5 min Then the blue supernatant was removed using pipette and leaving blue pellet in the tube. The blue pellet was washed using ethyl acetate several times until the supernatant became colorless. The pellet was then redissolved in 1 mL of acetone to measure the absorbance at 623 nm.

**Characterization.** Particle size and distribution (poly-dispersity index, PDI) were determined by dynamic light scattering (DLS) using a Microtrac NANO-flex. Zeta potential of particle dispersion were measured by Stabino PMX 400. FTIR absorbance peaks were acquired using Bruker Tensor 27 FTIR spectrometer and NP dispersion and solution were dropped on KBr pellets and dried at 50 °C. UV/Vis absorption spectra for F127 qualification was carried out with a UV–Vis–NIR spectrometer (Lambda 950, from Perkin Elmer). The photoluminescence spectra were measured using a JASCO spectrofluorometer FP-8500. SEM images were taken using a FEI Sirion SEM at 3 kV accelerating voltage. All samples were coated with 3 nm Pt before performing SEM measurements. AFM measurements were performed with a Nanosurf Easy Scan 2 in contact mode. Near-edge X-ray absorption fine structure (NEXAFS) spectroscopy have been recorded at the PolLux beamline at the Swiss Light Source[66]. The samples were raster-scanned through the focal spot of a nickel zone plate with an outermost zone width of 25 nm (yielding approx. 30 nm lateral resolution) with a pixel resolution of 10 nm.

**GIWAXS.** The GIWAXS diffractograms were collected with the highly customized Versatile Advanced X-ray Scattering instrument Erlangen (VAXSTER) at the Institute for Crystallography and Structural Physics (ICSP, FAU, Germany). The instrument is equipped with a 9.24 keV MetalJet D2 70 kV X-ray source (EXCILLUM, Sweden), a focusing 150 mm Montel optics (INCOATEC, Germany) and a fully evacuated beam path and sample stage. Two beam-shaping double-slits were used, with the last double-slit systems being low scattering silicon slits (JJXray/SAXSLAB, Denmark). Aperture sizes of the double-slit systems were $0.7 \times 0.7$ mm$^2$ and $0.462 \times 0.462$ mm$^2$, respectively. A 2D hybrid-pixel Pilatus 300 K detector (Dectris, Switzerland) was used to collect the scattered radiation. The incidence angle was set to 0.17°, which is in between the critical angles of the spin-coated P3HT:IDTBR layer and the Silicon substrates. This angle is used here to enhance the scattered intensity by maximization of the scattering sample volume and minimizing the background scattering from the substrate. The detector-to-sample distance (SDD) was calibrated with a silver behenate standard and was set to 169 mm. Data was reduced using the software dpdak[67] and was corrected for variations in sample thickness, of the X-ray beam and exposure times, with the expection of azimuthal plots being only corrected for time and flux. The software BornAgain (C. Durniak, M. Ganeva, G. Pospelov, W. Van Herck, J. Wuttke (2015), BornAgain—Software for simulating and fitting X-ray and neutron small-angle scattering at grazing incidence, version 1.9, http://www.bornagainproject.org) was used to depict the 2D patterns in $Q_y$ – $Q_z$ coordinate system. WAXS and SAXS experiments were performed at the same instrument as the GIWXS/GISAXS. The suspensions were in a glass capillary. The same capillary was used for the three suspensions. After SAXS/WAXS measurement the same capillary was measured with water. The measurement with water was subtracted. For absolute calibration a glassy carbon standard was used. The sample to detector distance was 1597.5 mm for SAXS and 178.285 mm for WAXS.

**Device.** In general, pre-structured indium tin oxide (ITO) substrates were cleaned with acetone and isopropyl alcohol in an ultrasonic bath for 10 min each. After drying, the substrates were coated with 40 nm of ZnO (Nanograde, N-10) by doctor-blading and then annealed at 85 °C for 5 min Aqueous ink with 80 mg mL$^{-1}$ P3HT/o-IDTBR, 50 mg mL$^{-1}$ PCE10/o-IDTBR, PBQ-QF/o-IDTBR or PBQ-QF/ITIC particle ink was then spin-coated at 1000 rpm onto ZnO surface under ambient atmosphere. The resultant film was annealed at 150 °C in glovebox for 10 min To complete the fabrication of the devices, 10 nm of MoO$_x$ and 100 nm of Ag were thermally evaporated through a mask (with a 10.4 mm$^2$ active area opening, the areas were determined with the optical microscope) under a vacuum of approximately $1 \times 10^{-6}$ mbar. Scan direction: forward; speed: 1 V s$^{-1}$; dwell time: 2 s. The current–voltage characteristics of the solar cells were measured under AM 1.5G irradiation on an OrielSollA Solar simulator (100 mW cm$^{-2}$). The light source was calibrated by using a silicon reference cell. More than four devices ($4 \times 6 = 24$ pixels) for each system were tested. All the devices were tested at 25 °C in air. An Enli Technology (Taiwan) IQE measurement system (QE-R) was used for measuring EQE and reflection of solar cells. The EQE calibrated $J$sc was listed in Supplementary Table 5. The IQE were determined following the equation: IQE = EQE/(1 − Reflection). The light intensity at each wavelength was calibrated with a standard single-crystal Si photovoltaic cell. All cells were tested under ambient air. The long-term photo-stability of P3HT:IDTBR devices were performed under continuous 1 sun illumination in a home-built chamber using a UV filter (cutoff 380 nm, the light spectrum is shown in Supplementary Fig. 29) filled with N$_2$.

**Photo-CELIV.** In photo-CELIV measurements, a 405 nm laser diode illuminated the solar cell. Current transients were recorded across the internal 50 Ω resistor of the oscilloscope (Agilent Technologies DSO-X 2024A). Then, in order to prevent carrier extraction or sweep out during the laser pulse and delay time, a fast electrical switch was used to isolate the solar cell. After the variable delay time, the solar cells was switched its connect to a function generator. A linear extraction ramp with 60 µs and a 2 V amplitude was applied. The linear extraction ramp was set to start with an offset matching the $V_{OC}$ of the solar cell. The maximum extracted current ($t_{max}$) can be determined after the measurements of photocurrent transients. The charge carrier mobility ($\mu$) is further calculated with the following equation:[51]

$$\mu = \begin{cases} \dfrac{2d^2}{3At_{max}^2 \left[1 + 0.36\frac{\Delta j}{j(0)}\right]} & \text{if } \Delta j \leq j(0) \\[2ex] \dfrac{2d^2}{3At_{max}^2} & \text{if } \Delta j \leq j(0) \end{cases} \tag{1}$$

where $d$ is the active layer thickness, $A$ is the voltage rise speed $A = dU/dt$, $U$ is the applied voltage, $\Delta j$ is the extracted current and $j(0)$ is the constant current.

In the case of bimolecular recombination, the charge density $n$ as a function of the combined time $t = (t_d + t_{max})$ ($t_d$: delay time) should follow equation:[52]

$$n(t) = \frac{n_0}{1 + n_0\beta t} \qquad (2)$$

where $n(t)$ is the charge density at time $t$ and $n_0$ is the initial charge density.

**TPV and CE.** In this measurement, a laser diode with a wavelength of 405 nm was used to keep the solar cells in approximately $Voc$ conditions. A waveform generator (Agilent 33500B) was used to drive the laser intensity and a highly linear photo-diode was used to measure the light intensity, allowing reproducible adjustments of the light intensity with an error below 0.5% over a range from 0.1 to 1 suns. A small perturbation was achieved by illuminating the solar cell by a second 405 nm laser diode. The intensity of this added short (50 ns) laser pulse was adjusted to keep the voltage perturbation below 10 mV, typically at 5 mV. After the pulse, the voltage decays back to its steady state value in a single exponential decay. A linear fit to a logarithmic plot of the voltage transient was used to determine the characteristic decay time, and returned the small perturbation charge carrier lifetime. In the case of charge extraction measurements, a 405 nm laser diode illuminated the solar cell for 200 μs to reach a constant $Voc$ with steady state conditions. At the end of this 200 μs illumination, a switch from open-circuit to short-circuit (50 Ω) conditions was applied to the solar cell within less than 50 ns. As described by Shuttle et al.[68], a correction was applied for the charge on the electrodes that results from the geometric capacity of the device.

Combining these two techniques, the charge carrier density ($n$) in relation to the carrier lifetime ($\tau$) is determined by the equation:[54]

$$\tau = \tau_0 \left(\frac{n_0}{n}\right)^\lambda \qquad (3)$$

where $\tau_0$ and $n_0$ are constant, and $\lambda$ is so called recombination exponent.

## Data availability

All data generated or analyzed during this study are included in the published article (and its Supplementary Information).

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

## Acknowledgements

C.X. and X.T. acknowledge the financial support from the China Scholarship Council (CSC). W.G., I.S., and T.U. gratefully acknowledge the funding of the Deutsche For-schungsgemeinschaft (DFG) through INST 90/825-1 FUGG, INST 90/751-1 FUGG, INST 90/827-1 FUGG, the "Cluster of Excellence Engineering of Advanced Materials (EAM)", the research training group GRK 1896 "In-Situ Microscopy with Electrons, X-rays and Scanning Probes" and the research unit FOR 1878 "Functional Molecular Structures on Complex Oxide Surfaces" and the German Federal Ministry of Education and Research (BMBF, project numbers: 05K16WEB, 05K16WE1). R.H.F acknowledges financial support from the BMBF (contract 05K16WED). N.L. gratefully acknowledges the financial support from the DFG research grant: BR 4031/13-1 and the Bavarian Ministry of Economic Affairs and Media, Energy, and Technology by funding the HI-ERN (IEK11) of FZ Jülich. C.J.B. gratefully acknowledges the financial support through the "Aufbruch Bayern" initiative of the state of Bavaria (EnCN and "Solar Factory of the Future"), the Bavarian Initiative "Solar Technologies go Hybrid" (SolTech), and the SFB 953 (DFG).

## Author contributions

C.X. and N.L. conceived and developed the idea. C.X. designed and coordinated the experiments, performed NP synthesis and characterization, performed device fabrication, and characterization and data analysis. T.H. and A.C. performed Photo-CELIV, TPV & CE and light intensity dependency measurement and analyzed the data. T.H. performed stability test of devices. X.T. performed SEM measurements and analyzed the data. W.G., I.S., and T.U. performed GIWAXS, WAXS and SAXS measurements and analyzed the data. M.B. and I.M. provide the materials for experiment. A.S. and R.H.F. performed NEXAFS and data analysis. N.L. and C.J.B. supervised the project. C.X., N.L., and C.J.B. wrote the manuscript with input from all co-authors.

## Additional information

**Competing interests:** The authors declare no competing interests.

