## [Peer Review File · Nature Communications]

Reviewers' comments:

Reviewer #1 (Remarks to the Author):

In this manuscript, the authors present evidences on the material-processing-property relationship that must be met in order to have efficient water-processed bulk-heterojunction organic solar cells (OSCs) based on non-fullerene acceptors nanoparticles (NPs). In this regard, they present an elegant and generic approach to processing NP-based OSCs from water with minimized surfactant-induced microstructure defects, achieving impressively high device performance of >5% and >7% for P3HT:IDTBR and PBQ-QF:ITIC OSCs, respectively. These impressively high performances are even comparable with those of processed with toxic halogenated solvents. They in-depth investigate the devices processed using different surfactants and processing conditions, and deliver important messages to the community regarding eco-friendly and reliable processing of OSCs from the cleanest solvent of water. GIWAXS experiment demonstrate that the NPs synthesized based on the concept show highly ordered big domains prior to thermal annealing process; and internal quantum efficiency and transport properties measurements show that while having big domains and minimized microstructure defects helps having high fill factor and photocurrent, leading to the impressively high performance. These observations represent a breakthrough that allows the rational design of solvent surfactants and processing conditions suitable for various donor polymers and small molecule acceptors. In my viewpoint this work meets the criteria of novelty and quality of Nature Communications, thus I would like to recommend its publication after addressing the following points:

- 1) The stability of NP inks, which is very much relevant to the industrial production, should be mentioned. Did the author observe distinct ink stability for different NP systems?
- 2) How about the thickness-dependent performance of P3HT:IDTBR NP solar cells? Is it possible to fabricate NP OSCs with thick active layers? Is the performance influenced by the size distribution of NPs.
- 3) The light intensity dependence of device performance has to be added and discussed.
- 4) It is required to provide the device stability data of other NP OSCs, in particular the most efficient system, PBQ-QF:ITIC.
- 5) As the authors claim several times that, both in Abstract and main text, "the observed water-processed devices with photovoltaic performances comparable to those processed from halogenated solvents and solvent mixtures", relevant efficiencies of devices composed of the same active layer that were processed with "halogenated solvents and solvent mixtures" should be provided in Figure 1a (Table S1), and relevant references should be cited accordingly, for example, line 292-295 of Page 10.

Reviewer #2 (Remarks to the Author):

This manuscript convincingly describes an original and successful approach to reduce the surfactant concentration in NP-BHJ solar cells. As a consequence the efficiency has also significantly been improved. The study is very thorough, including many additional analytical techniques and several material combinations. Processing from water is a relevant improvement over other solvents and this work represents a major step forward in that direction. Therefore I recommend this to be published in Nature communications. Below I have some comments/question that could help to further improve the quality of this work.

- It seems rather counterintuitive that the csNP films show a high degree of anisotropy, as evidenced by the in- and out-of-plane X ray diffraction patterns. Please comment.

- In my opinion Fig4c has no added value. It just a different representation of the data in Fig 4a, and every observation based on this graph can also be made looking at Fig4a
- Personally, I find the short paragraph (lines 228-231) about PV efficiency of P3HT/ICBA a bit confusing because all data described before that point dealt with P3HT/IDTBR.
- Authors calculate with a suggested high accuracy the exciton harvesting efficiency. I question this, because it is not taken into account (actually cannot) that the PL quantum yield of a material can depend greatly on the morphology/crystallinity, which is very different for the pure materials vs. blended materials. This is certainly the case for P3HT. So any quantification is sketchy.
- Fig5h suggests that the material domains in the csNPs are larger than in NPs with surfactants. I do not see any data that confirms this
- Authors demonstrate that the improved performance for P3HT/IDTBR csNPs relates to improved carrier collection efficiency, rather than carrier generation efficiency. However, looking at the other material combinations that have been incorporated to demonstrate the universality of the approach, it seems to me that these all behave differently. The JV curves of these material combinations (Fig 5b-d) all show low fill factors and high field dependent carrier collection, also for the scNPs, which have similar slopes at J_{sc} as the SDS-NP blends, indicating similar carrier collection problems. This raises the question if the behavior of P3HT is representative at all.
- Please comment on the stability of the washed csNP dispersions
- The experimental methods do not mention the method of deposition of the device films (ZnO and NPs). Please indicate. Coating from water typically does not improve film formation. Please comment on this.
- Even though several material combinations are included in the manuscript, I imagine the method is not suitable for all materials. Are there any combination tested and not (yet) proven successful? I encourage authors to mention these as well.

Reviewer #3 (Remarks to the Author):

This paper reports the application of poloxamers as surfactants, with temperature dependent micelle formation, in the fabrication of nanoparticulate organic photovoltaic (NP-OPV) devices. Using these materials in conjunction with highly performing donor-acceptor blends, the authors report the highest efficiencies for NP-OPV devices to date. The authors' conclusions are well-supported by extensive experimental data from a wide range of experimental techniques. The paper makes a new and substantial contribution to the field of NP-OPV research and warrants publication in Nature Communications once the following recommended revisions are addressed.

1. Page 3 L64-66: The last sentence in the paragraph referring to inkjet printing of OPV at home is not really warranted at this stage and should be removed.
2. Page 8 L234: The word "amazingly" should be removed.
3. Page 9 L237-239: The authors state that the centrifugal washing played a critical role in the NP-OPV fabrication. They need to elaborate on this statement and explain how they understand the how the washing process determines device performance.
4. Page 9 L261: Correct "charrier" to "carrier"
5. Page 9 L267: Replace "exhibits" with "shows".
6. Page 9/10: The discussion around recombination dynamics is confusing. The authors argue that the recombination order ($R > 2$) reflects increased trapping and thus $R_{SDS} > R_{cs-NP}$ is consistent with residual SDS creating additional trap states. However, they do not discuss explicitly the fact that $R_{BHJ} > R_{SDS} > R_{cs-NP}$ implying that the greatest trap density occurs for the BHJ system and yet the performance of the BHJ system is consistently intermediate between the SDS and cs-NP systems as measured by all of the

characterisation techniques. The authors need to discuss and clarify.

7. Page 10 L300-301: The degradation data shows that the NP-OPV devices degrade without a characteristic “burn-in” phase. Can the authors explicitly comment on this observation and its implication for the degradation mechanisms in these devices?

8. Page 10 L292 – 295: The authors state that the performance enhancement in cs-NP-OPV devices can be mainly attributed to decreased energetic disorder caused by residual surfactant. Indeed, the main conclusion of the paper is the removal of excess surfactant is the dominant improvement mechanism, through the elimination of trap sites. However, what is not clear from the paper is whether this effect could also be due to an improvement in film morphology driven by the reduced surfactant concentration. In other words, while it is clear that removing excess surfactant improves performance, is that improvement due to: (a) residual surfactant not impeding charge transport or, (b) the reduced surfactant concentration driving a new morphological or structural state of the film, or (c) a combination of (a) and (b). It is clear from the paper that the authors believe that (a) is the dominant effect (given the evidence that they have), however, what is lacking from the paper is a really detailed structural study that might identify if there are substantial changes in film structure. In the absence of this study, the authors either need to provide further justification/evidence for their conclusion that (a) is dominant or to alter their conclusions to note that (b) or (c) are also possibilities.

9. Page 10 L322-324: The authors state that the residual surfactant in non-annealed SDS-NP films deteriorates crystallinity. While it is true that there is residual surfactant and that the crystallinity is reduced it is not clear that there is evidence for a causal link between the two observations. What would be the mechanism? The authors need to provide further justification for this statement.

10. Finally, given that the paper is predicated on the need for large scale printing of OPVs it would be good for the authors to discuss briefly how the use of these poloxamer materials might be implemented for an actual printing process and whether these materials are already used at scale. Is this approach viable for large scale production?

Response to reviewers' comments:

Reviewer #1:

In this manuscript, the authors present evidences on the material-processing-property relationship that must be met in order to have efficient water-processed bulk-heterojunction organic solar cells (OSCs) based on non-fullerene acceptors nanoparticles (NPs). In this regard, they present an elegant and generic approach to processing NP-based OSCs from water with minimized surfactant-induced microstructure defects, achieving impressively high device performance of >5% and >7% for P3HT:IDTBR and PBQ-QF:ITIC OSCs, respectively. These impressively high performances are even comparable with those of processed with toxic halogenated solvents. They in-depth investigate the devices processed using different surfactants and processing conditions, and deliver important messages to the community regarding eco-friendly and reliable processing of OSCs from the cleanest solvent of water. GIWAXS experiment demonstrate that the NPs synthesized based on the concept show highly ordered big domains prior to thermal annealing process; and internal quantum efficiency and transport properties measurements show that while having big domains and minimized microstructure defects helps having high fill factor and photocurrent, leading to the impressively high performance. These observations represent a breakthrough that allows the rational design of solvent surfactants and processing conditions suitable for various donor polymers and small molecule acceptors. In my viewpoint this work meets the criteria of novelty and quality of Nature Communications, thus I would like to recommend its publication after addressing the following points:

- 1) The stability of NP inks, which is very much relevant to the industrial production, should be mentioned. Did the author observe distinct ink stability for different NP systems?

Reply: We are grateful to the referee for the very positive evaluation and comments! We have noticed that the prepared nanoparticle inks were quite stable according to the DLS data. We therefore added the DLS data of cs-NP dispersion stored at around 0 °C for 3 months to the manuscript (Fig. 2e). The discussion was added to page 6 in section “**Synthesis of NFA-based surfactant-stripped nanoparticles**” as follows: “The constancy of averaged size of cs-NPs (Fig. 2e) demonstrates that the cs-NPs dispersed in water are highly stable and no significant occurrence of Ostwald ripening and only minor sedimentation was observed after three months

storage at $\sim 0^\circ\text{C}$ (Dimitrov, A. S. & Nagayama, K. Continuous Convective Assembling of Fine Particles into Two-Dimensional Arrays on Solid Surfaces. *Langmuir* 12, 1303–1311 (1996).)”

e, Tyndall effect observed from P3HT:IDTBR cs-NPs dispersion and size distribution of cs-NPs measured by DLS before and after storage in a freezer at approximately 0°C for 3 months.

2) How about the thickness-dependent performance of P3HT:IDTBR NP solar cells? Is it possible to fabricate NP OSCs with thick active layers? Is the performance influenced by the size distribution of NPs.

Reply: We performed the thickness-dependent performance of P3HT:IDTBR NP solar cells and summarized the corresponding j - V characteristics in the following figure (Supplementary **Fig. S15**). The discussion was added to page 9 in the section “**Characterization of cs-NP-based organic solar cells**” as follows: “Furthermore, the PCE of cs-NP device was almost insensitive to the active layer thickness up to 300 nm (**Fig. S15**).”

Figure S15. **Thickness dependence of P3HT:IDTBR cs-NP solar cells.** a, J - V characteristics and b, PCE as a function of active layer thickness of cs-NP devices.

3) The light intensity dependence of device performance has to be added and discussed.

Reply: Please see the figure below. The discussion was added to page 10 in the section “**Characterization of cs-NP-based organic solar cells**” as follows: “Light intensity-dependent V_{OC} measurements of cs-NP device suggest slightly reduced first-order recombination compared to the control devices^{59–61} (**Fig. S22**)”

Figure S22. **a**, V_{OC} and **b**, J_{SC} of the three P3HT:IDTBR solar cells as a function of light intensity.

4) It is required to provide the device stability data of other NP OSCs, in particular the most efficient system, PBQ-QF:ITIC.

Reply: The figure below shows the stability results of the PCE10:IDTBR, PBQ-QF:IDTBR and PBQ-QF:ITIC cs-NP solar cells. The discussion has been added to page 11 in section “**University of surfactant-stripping technique**” as follows: “The stability data of the three systems under continuous one sun illumination is shown in **Fig. S28**. Both IDTBR-based solar cells presented PCE losses less than 5% after 1000 hours illumination, while a “burn-in” degradation ($\sim 15\%$) was observed for PBQ-QF:ITIC-based cs-NP solar cells.”

Figure S28. Normalized PCE evolution of **a**, PCE10:IDTBR, **c**, PBQ-QF:IDTBR and **d**, PBQ-QF:ITIC cs-NP solar cells measured under continuous one sun illumination in N_2 .

5) As the authors claim several times that, both in Abstract and main text, “the observed water-processed devices with photovoltaic performances comparable to those processed from halogenated solvents and solvent mixtures”, relevant efficiencies of devices composed of the same active layer that were processed with “halogenated solvents and solvent mixtures” should be provided in Figure 1a (Table S1), and relevant references should be cited accordingly, for example, line 292-295 of Page 10.

Reply: The PCEs reported in literature were reviewed and summarized in **Table S1** as shown below. The corresponding references were added to the supplementary information. The efficiency comparison between NP and BHJ solar cells was summarized in **Fig. S1**.

Figure S1. Efficiency evolution of NP dispersion-processed organic solar cells and the relevant efficiencies of solution-processed BHJ solar cells with the same active layers.

Table S1. Reported PCEs for water/alcohol NP dispersion processed organic solar cells and their relevant PCEs of solar cells with the same active layers processed by toxic solvents.

Year	Materials	Good solvent	Bad solvent	Surfactant	Best PCE [%]	Reference	PCE from toxic processing [%]	Solvent	Reference
2003	F8BT:PFB	CF ^a	Water	SDS	0.004	S1	0.2	CF	S1
2011	PSBTBT:PCBM	CF	Water	SDS	0.55	S2	5.1	CB ^b	S3
2012	P3HT:PCBM	CF	Water	SDS	0.29	S4	3.13	CB	S4
2012	F8BT:PFB	CF	Water	SDS	0.39	S5	0.2	CF	S1
2013	P3HT:PCBM	CF	Water	SDS	1.31	S6	3.13	CB	S4
2013	P3HT:ICBA	CF	Water	SDS	2.50	S7	5.44	DCB ^c	S8
2014	P3HT:PCBM	CF	Ethanol	-	1.09	S9	3.13	CB	S4
2014	PDPPTNT:PC ₇₁ BM	CF	Water	SDS	1.99	S10	3.6	CF+DCB	S11
2014	P3HT:PCBM	CF	Water	SDS	2.15	S12	3.13	CB	S4
2014	P3HT:ICBA	CF	Ethanol	-	3.5	S13	5.44	DCB	S8
2015	P3HT:PCBM	CF	Water	SDS	1.16	S11	3.13	CB	S4
2015	P(TBT-DPP):ICBA	CF	Water	SDS	2.63	S15	4.75	CF+DCB	S16
2016	TQ1: PC ₇₁ BM	CF	Water	SDS	2.54	S17	6.0	DCB	S18
2016	P3HT:ICBA	CF	Ethanol	-	4.3	S19	5.44	DCB	S8
2016	PBDTPD:PC ₇₁ BM	CB	Water	SDS	3.8	S20	7.9	CB+CN ^d	S21
2017	PCDTBT: PC ₇₁ BM	CB	Water	SDS	1.9	S22	6.33	CB+DCB	S23
2017	PDPP5T: PC ₇₁ BM	CF	Water	SDS	2.36	S24	5.74	CF+DCB	S25
2018	PCDTBT:PC ₇₁ BM	THF ^e	Water	-	0.33	S26	6.33	CB+DCB	S23
2018	PDPP5T-2: PC ₇₁ BM	CF	Water	SDS	3.38	S25	5.74	CF+DCB	S25
2018	P3HT:ICBA	CF	Ethanol	-	4.52	S27	5.44	DCB	S8
2018	PTNT:PC ₇₁ BM	xylene	Water	SDS	1.65	S28	4.6	DCB	S29
2018	P3HT:o-IDTBR	THF	Water	F127	5.23	This work	6.3	DCB	S30
2018	PCE10:o-IDTBR	THF	Water	F127	5.19	This work	9.57	DCB	S31
2018	PBQ-QF:o-IDTBR	THF	Water	F127	6.52	This work	-	-	-
2018	PBQ-QF:ITIC	THF	Water	F127	7.50	This work	8.90	THF+IPA ^f	S32

^a Chloroform. ^b Chlorobenzene. ^c ortho-Dichlorobenzene. ^d 1-chloronaphthalene ^e Tetrahydrofuran. ^f Isopropanol

Reviewer #2:

This manuscript convincingly describes an original and successful approach to reduce the surfactant concentration in NP-BHJ solar cells. As a consequence the efficiency has also significantly been improved. The study is very thorough, including many additional analytical techniques and several material combinations. Processing from water is a relevant improvement over other solvents and this work represents a major step forward in that direction. Therefore I recommend this to be published in Nature communications. Below I have some comments/question that could help to further improve the quality of this work.

- It seems rather counterintuitive that the csNP films show a high degree of anisotropy, as evidenced by the in- and out-of-plane X ray diffraction patterns. Please comment.

Reply: We greatly thank the referee for the positive evaluation of our work. To clarify the concern raised by the referee, we performed WAXS and SAXS measurements on various NP dispersions, and AFM measurement on individual nanoparticles spin-coated on silicon substrate. The experimental data suggest that this anisotropy is originated from the deformation of NPs during film formation. A model about the NP film formation was further suggested:

“Transmission X-ray scattering. Three dispersions, SDS-stabilized NP, cs-NP after 2 times centrifugal washing and cs-NP after 5 times washing were measured by transmission WAXS and SAXS. An intensive peak at 3.6 nm^{-1} was observed for all three dispersions. The SAXS patterns of the three samples studied were found to be very similar as well. Fitting the experimental data by applying a model of isolated compact and homogeneous spheres with a Gaussian shaped size distribution, the data can be well reproduced. The diameters of NPs in the three samples are similar. We therefore conclude that the nanoparticles in solution are (i) crystalline and (ii) spherical. Next, we investigated the shape of single NP after spin coating by AFM

Figure S11. Transmission WAXS and SAXS profiles of SDS stabilized P3HT:IDTBR NP dispersion as well as

cs-NP dispersion after 2 and 5 times centrifugal washes.

NP deformation. Diluted P3HT:IDTBR cs-NP dispersions (1 mg/mL) were spin-coated (at 1000 rpm) on a Si substrate. As shown in Fig. S11, the selected 9 particles all exhibited the shape of an ellipsoid with a width to height ratio of ~ 2 . This observation suggests that the NPs (soft polymer:NFA colloids) are deforming during film formation, which results into a different in-plane vs out-of-plane aspect ratio. (ref S14, Pedersen, E. B. L. *et al.* Structure and crystallinity of water dispersible photoactive nanoparticles for organic solar cells. *J. Mater. Chem. A* **3**, 17022–17031 (2015).)

Figure S12. **a**, AFM height image of isolated P3HT:IDTBR NPs spin-coated on silicon substrate and **b**, the extracted height profiles of the numbered 9 particles.”

The discussion has been added to section “**Water-processing of cs-NP film**” : “NP dispersions do not show preferential orientation or anisotropic differences in crystallinity as evidenced by

transmission wide-angle X-ray scattering (WAXS) and transmission small-angle X-ray scattering (SAXS) (described in Supplementary information, section “**Transmission X-ray scattering**”). However, AFM measurements evidence deformation of nanoparticles during film formation (Fig S12) (described in Supplementary information, section “**NP deformation**”), suggesting geometrical rearrangement of the single particles during during spin-coating or drying.”

In addition, the discussion about the model has been added to this section: “Summarizing the above observations, we suggest the following model to rationalize the shape of NPs after film deposition in **Fig. S14**. NP touching the substrate surface during spin coating become deformed to considerable shear forces, resulting from the difference in adhesive and centrifugal forces.. Residual surfactants on the NP surface hamper solid state recrystallization between single nanoparticles. On the other hand, surfactant free NPs can undergo reorganization and eventually recrystallization at their grain boundaries. Overall, we highlight that cs-NP films suffer significantly less disturbance from the surfactant and exhibit significantly higher crystallinity than films from SDS-NP.”

Figure S14. Schematic overview of the structure and crystallinity of P3HT:IDTBR NP with and without surfactant during film deposition. The NPs are smeared out along the surface and deformed into a film during spin-coating. The residual surfactant on the NP surface hampers the recrystallization of polymer and NFA during NP deformation process. For the NP after surfactant-stripping, NPs prefer to merge with each other and lead to a film with high crystallinity.

Based on those results, we suggest that the anisotropy in NPs is probably formed during the NP deposition.

- In my opinion Fig4c has no added value. It just a different representation of the data in Fig 4a, and every observation based on this graph can also be made looking at Fig4a

Reply: We fully agree that the photocurrent behavior is similar to the J-V characteristics in Fig. 4a. However, the photocurrent density J_{ph} in Fig. 4c is the difference between the light and dark current density, and represents the extracted charges under different internal voltages. Therefore, Fig. 4c exhibits the charge generation behavior in those NP devices excluding the impact of dark current. Please see the table listed below (Table 2 in the manuscript). The three systems exhibited very similar saturated photocurrents, which cannot be that clearly observed in Fig. 4a. This result clearly indicates that charge generation does not dominate J_{sc} losses in these two control device systems, which can not been seen from Fig4a. Therefore, we suggest to keep this figure in the main text for better comparison.

The discussion was revised as follows: “The maximum charge carrier generation rate G_{max} (Table 2) of those three systems indicates that the charge generation does not dominate the losses in those two control devices. The saturation current density is defined by $J_{sat} = qG_{max}L$, where q is the elementary charge and L is the active layer thickness (Table 1).”

	J_{sat} [mA cm ⁻²]	G_{max} [cm ⁻³ s ⁻¹]	IQE ^a [%]	$\eta_{eh,D}$ [%]	$\eta_{eh,A}$ [%]	η_{cc}^a [%]	μ [cm ² V s ⁻¹]	β [cm ³ s ⁻¹]	τ [μs]	λ
THF	11.97	3.44×10^{-21}	45.2	74.9	68.8	65.5	1.14×10^{-4}	2.61×10^{-13}	1.46	2.95
Water (SDS)	11.25	3.35×10^{-21}	42.7	70.6	71.7	59.6	6.88×10^{-5}	1.89×10^{-13}	1.45	2.36
Water (F127)	12.01	3.26×10^{-21}	53.8	76.3	72.4	74.2	4.04×10^{-4}	1.72×10^{-13}	1.81	1.89

- Personally, I find the short paragraph (lines 228-231) about PV efficiency of P3HT/ICBA a bit confusing because all data described before that point dealt with P3HT/IDTBR.

Reply: According to this suggestion, the discussion about the ICBA system was removed.

-Authors calculate with a suggested high accuracy the exciton harvesting efficiency. I question this, because it is not taken into account (actually cannot) that the PL quantum yield of a material can depend greatly on the morphology/crystallinity, which is very different for the pure materials

vs. blended materials. This is certainly the case for P3HT. So any quantification is sketchy.

Reply: We fully agree that the crystallinity of pure P3HT is different from that of P3HT:NFA films. To estimate the absolute exciton harvesting efficiency, the changes of P3HT morphology/crystallinity have to be taken into consideration.

The uncertainty of this calculation has been elaborated in section “**Charge collection efficiency calculation.**” in Supplementary information as follows: “In this model, two assumptions should be satisfied:

- (1) The crystallinity of donor and acceptor in all the films are identical.
- (2) The quantum efficiency of PL is identical in any wavelength.

Due to the uncertain factors shown above, there would be considerable discrepancy between the calculated values and exact values.”

In addition, the discussion for this part in section “**Characterization of cs-NP-based organic solar cells**” was revised as follows: “Another method was introduced to understand the loss mechanism in THF and water (SDS) processed devices. The exciton harvesting efficiency (η_{eh}) and charge collection efficiency (η_{cc}) were calculated based on the results of PL and IQE results (described in the Supplementary information, section “**Charge collection efficiency calculation**”). The cs-NP device exhibits an assumption of both slightly higher η_{eh} and η_{cc} as compared to both control devices, indicating a relatively efficient exciton splitting and charge collection in cs-NP system.”. At the same time, the plot of charge collection efficiency in Fig. 4e has been moved to Fig. S20.

- Fig5h suggests that the material domains in the csNPs are larger than in NPs with surfactants. I do not see any data that confirms this

Reply: We fully agree that the larger domain in **Fig.5h** might be not accurate. We revised the figure as follows:

Figure 5. **h**, Schematic diagram of NP (SDS) and cs-NP structures as well as morphology and charge transport in their corresponding NP devices.

- Authors demonstrate that the improved performance for P3HT/IDTBR csNPs relates to improved carrier collection efficiency, rather than carrier generation efficiency. However, looking at the other material combinations that have been incorporated to demonstrate the universality of the approach, it seems to me that these all behave differently. The JV curves of these material combinations (Fig 5b-d) all show low fill factors and high field dependent carrier collection, also for the scNPs, which have similar slopes at J_{sc} as the SDS-NP blends, indicating similar carrier collection problems. This raises the question if the behavior of P3HT is representative at all.

Reply: The universality concept discussed in the manuscript suggests that the surfactant-stripping technique typically results in enhanced film crystallinity and solar cell performance as compared to traditional approaches in all four donor/acceptor systems, such as SDS-based NPs. We completely agree that the NP solar cells still exhibit relatively lower FF compared to corresponding optimized BHJ solar cells, in particular for the low-bandgap polymer systems. It is true that the cs-NP has overcome the low FF and the charge collection problems for SDS NP solar cells in P3HT system, but this is obtained after systematic device optimizations. However, the other systems were not optimized as much as P3HT, because we don't have enough material to run many tests, such as the particle size, annealing temperature, and film thickness and so on. We are confident that we should be able to improve morphology and thus collection efficiency by optimizing those factors.

The discussion has been added to section “**University of surfactant-stripping technique**” as follows: “Although the FFs are a little inferior than those of the optimized P3HT system, further morphology modifications on NP size, film thickness and annealing temperature would

overcome the charge collection problems in those low-bandgap systems.”

- Please comment on the stability of the washed csNP dispersions

Reply: We are grateful to the referee for the very positive evaluation and comments! We have noticed that the prepared nanoparticle inks were quite stable according to the DLS data. We therefore added the DLS data of cs-NP dispersion stored at around 0 °C for 3 months to the manuscript (Fig. 2e). The discussion was added to page 6 in section “**Synthesis of NFA-based surfactant-stripped nanoparticles**” as follows: “The constancy of averaged size of cs-NPs (Fig. 2e) demonstrates that the cs-NPs dispersed in water are highly stable and no significant occurrence of Ostwald ripening and only minor sedimentation was observed after three months storage at ~0 °C (Dimitrov, A. S. & Nagayama, K. Continuous Convective Assembling of Fine Particles into Two-Dimensional Arrays on Solid Surfaces. *Langmuir* 12, 1303–1311 (1996).)”

e, Tyndall effect observed from P3HT;IDTBR cs-NPs dispersion and size distribution of NPs before and after storage by DLS. The dispersion sample was kept in the freezer at approximately 0 °C for 3 months.

- The experimental methods do not mention the method of deposition of the device films (ZnO and NPs). Please indicate. Coating from water typically does not improve film formation. Please comment on this.

Reply: The processing method was added to the “Device” section: “After drying, the substrates were coated with 40 nm of ZnO (Nanograde, N-10) by doctor-blading and then annealed at 85 °C for 5 min. Aqueous ink with 80 mg/mL P3HT/o-IDTBR, 50 mg/mL PCE10/o-IDTBR, PBQ-QF/o-IDTBR or PBQ-QF/ITIC particle ink was then spin-coated at 1000 rpm onto ZnO

surface under ambient atmosphere.”

We agree with the referee that the NP film formation is not perfect. The comment has been added to the section “**Water-processing of cs-NP film**” in page 6: “Typically, the water processing of NP does not improve the film formation. Thus, thermal annealing was applied to overcome the inhomogeneity of as-cast NP layers”.

- Even though several material combinations are included in the manuscript, I imagine the method is not suitable for all materials. Are there any combination tested and not (yet) proven successful? I encourage authors to mention these as well.

Reply: We fully agree that this is an important issue for this new approach. We have tested several other material systems, including some systems with PCBM and some other highly efficient polymers. The solubility in THF turned out to be the limiting factor for this strategy. The fullerene and some polymers such as PffBT4T-2OD (PCE11) have very low solubility in THF. Consequently, we were unable to form stable and highly concentrated NPs using this approach.

The discussion was added to the end of “**University of surfactant-stripping technique**” section as follows: “An impressive device performance and stability was achieved via surfactant-stripping for these systems. However, we noticed that this technique might be incompatible with some material systems, for instance materials with very low solubility in THF and materials suffered from strong aggregations during NP synthesis. Therefore, the choice of materials with high solubility in THF is one essential consideration for a successful cs-NP synthesis and device fabrication.”

Reviewer #3:

This paper reports the application of poloxamers as surfactants, with temperature dependent micelle formation, in the fabrication of nanoparticulate organic photovoltaic (NP-OPV) devices. Using these materials in conjunction with highly performing donor-acceptor blends, the authors

report the highest efficiencies for NP-OPV devices to date. The authors' conclusions are well-supported by extensive experimental data from a wide range of experimental techniques. The paper makes a new and substantial contribution to the field of NP-OPV research and warrants publication in Nature Communications once the following recommended revisions are addressed.

1. Page 3 L64-66: The last sentence in the paragraph referring to inkjet printing of OPV at home is not really warranted at this stage and should be removed.

Reply: We agree with the reviewer that this sentence might be inappropriate in this paragraph. The following sentence was removed in the revised manuscript.

“Most importantly, the development of a water-based OPV ink enables the fabrication of customized devices utilizing a desktop inkjet printer at home, which could significantly facilitate the application of OPV in daily-life scenarios.”

2. Page 8 L234: The word “amazingly” should be removed.

Reply: The word was removed in the revised manuscript.

3. Page 9 L237-239: The authors state that the centrifugal washing played a critical role in the NP-OPV fabrication. They need to elaborate on this statement and explain how they understand the how the washing process determines device performance.

Reply: We added the following sentence in section “**Characterization of cs-NP-based organic solar cells**” to explain how the washing step affects the device performance.

“As shown in **Fig. S16** the devices processed by dispersion with less washing steps remain more residual surfactant, resulting in losses in FFs and PCEs.”

4. Page 9 L261: Correct “charrier” to “carrier”

Reply: This word was corrected.

5. Page 9 L267: Replace “exhibits” with “shows”.

Reply: This word was corrected.

6. Page 9/10: The discussion around recombination dynamics is confusing. The authors argue that the recombination order ($R > 2$) reflects increased trapping and thus $R_{SDS} > R_{cs-NP}$ is consistent with residual SDS creating additional trap states. However, they do not discuss explicitly the fact that $R_{BHJ} > R_{SDS} > R_{cs-NP}$ implying that the greatest trap density occurs for the BHJ system and yet the performance of the BHJ system is consistently intermediate between the SDS and cs-NP systems as measured by all of the characterisation techniques. The authors need to discuss and clarify.

Reply: We agree that the BHJ device indeed has the highest value of R , which indicates the strong trapping effects in THF-processed P3HT:IDTBR films. The fast decay of charge carriers as a function of charge density would cause a loss in V_{oc} . That is reason that the averaged V_{oc} value in the THF-processed film is slightly lower than the other two systems (Table 1). However, the trapping effect is not the only loss mechanism in this system. As we can see from the photo-CELIV result in Fig. 4f, the charge carrier mobility of BHJ device is intermediate between SDS and cs-NP systems, leading to a moderate J_{sc} of 8.38 mA cm^{-2} and a FF of $\sim 48\%$. Although the R value of the SDS system is lower than that of the BHJ system, the ultimate device performance is also determines by other factors, including charge carrier mobility.

7. Page 10 L300-301: The degradation data shows that the NP-OPV devices degrade without a characteristic “burn-in” phase. Can the authors explicitly comment on this observation and its implication for the degradation mechanisms in these devices?

Reply: “Burn-in” degradation typically depends on the nature of material systems, in particular on their microstructure morphology and the miscibility between donor and acceptor. According to literature, the P3HT:IDTBR does not show “burn-in” degradation [Gasparini, N. *et al.* Burn-in Free Nonfullerene-Based Organic Solar Cells. *Adv. Energy Mater.* **7**, 1700770 (2017).], which can be attributed to the highly crystalline nature of BHJ morphology after thermal annealing as well as the good miscibility between P3HT and IDTBR. To clarify, the following sentence was added to this paragraph:

“Similar to the P3HT:IDTBR solar cells processed from halogenated solvents, no photo-induced “burn-in” losses were observed in water-processed NP solar cells,

8. Page 10 L292 – 295: The authors state that the performance enhancement in cs-NP-OPV devices can be mainly attributed to decreased energetic disorder caused by residual surfactant. Indeed, the main conclusion of the paper is the removal of excess surfactant is the dominant improvement mechanism, through the elimination of trap sites. However, what is not clear from the paper is whether this effect could also be due to an improvement in film morphology driven by the reduced surfactant concentration. In other words, while it is clear that removing excess surfactant improves performance, is that improvement due to: (a) residual surfactant not impeding charge transport or, (b) the reduced surfactant concentration driving a new morphological or structural state of the film, or (c) a combination of (a) and (b). It is clear from the paper that the authors believe that (a) is the dominant effect (given the evidence that they have), however, what is lacking from the paper is a really detailed structural study that might identify if there are substantial changes in film structure. In the absence of this study, the authors either need to provide further justification/evidence for their conclusion that (a) is dominant or to alter their conclusions to note that (b) or (c) are also possibilities.

Reply: We fully agree with the referee that we need to clarify this issue. As shown in the figure below, the optical microscopy clearly shows that residual F127 significantly perturbs film formation . In addition, we have performed GIWAXS measurements on cs-NP film processed from NP dispersions after 2 times centrifugal washing (more F127 remained in NP films). As the cs-NPs after 2 times washing only contain 4% of F127, the (100), (200) and (300) peaks are slightly lower than the surfactant-stripped NP film from out-of-plane cuts. Based on this result, we added the following sentence to the manuscript in the section “**Water-processing of cs-NP film**”:

“**Fig. S8** exhibits that the amount of residual F127 significantly affects microstructure formation of the deposited thin films.” and “Meanwhile, the relatively weakened diffraction peaks from out-of-plane cuts has also been observed in F127-stabilized NP film with incomplete washing process (**Fig. S10**).”.

Figure S8. Optical microscope images of as cast P3HT:IDTBR films processed from cs-NP dispersion (a) without, after (b) 1 time, (c) 2 times, (d) 3 times (e) 4 times and (f) 5 times centrifugal washes.

Figure S10. GIWAXS profiles of as cast P3HT:IDTBR films from cs-NPs after 2 times and 5 times centrifugal washes, respectively, collected from **a**, out-of-plane cuts and **b**, in-plane cuts. As shown in Fig. S8, the NP films with less than once washing step are not homogenous, which are not appropriate for GIWAXS measurement. The film with 2 times washes contains 4% of F127. The relatively low (100), (200) and (300) peaks from out-of-plane cuts indicates that the residual surfactant would deteriorate the crystallinity of NP film.

Based on those measurement, it is clear that the residual surfactant would induce a distinctly different film structure. In this case, we conclude that the residual surfactant would (a) deteriorate the charge transport and also (b) cause lower crystalline films. Both factors affect the performance of the NP device.

According to the referee's comments, the sentence in L292-293 was revised to "To summarise, the significant enhancement of FF and J_{SC} in cs-NP devices is mainly attributed to the increased

charge carrier mobility and the decreased microstructural disorder caused by residual stabilizers and incomplete crystalline”.

9. Page 10 L322-324: The authors state that the residual surfactant in non-annealed SDS-NP films deteriorates crystallinity. While it is true that there is residual surfactant and that the crystallinity is reduced it is not clear that there is evidence for a causal link between the two observations. What would be the mechanism? The authors need to provide further justification for this statement.

Reply: As discussed in the previous statement, the GIWAXS in **Fig. S10** and the optical microscope images in **Fig. S8** exhibit that residual surfactant F127 deteriorates crystallinity. In addition, we introduced a model for elaborating the mechanism. As shown in the figure below:

Figure S14. Schematic overview of the structure and crystallinity of P3HT:IDTBR NPs with and without surfactant during film deposition. The NPs are smeared out along the surface and deformed into a film during spin-coating. The residual surfactant on the NP surface hampers the recrystallization of polymer and NFA during NP deformation process. For the NP after surfactant-stripping, NPs prefer to merge with each other and lead to a film with high crystallinity.

The elaboration has been added to section “**Water-processing of cs-NP film**”: “Summarizing the above observations, we suggest the following model to rationalize the shape of NPs after film deposition in **Fig. S14**. NP touching the substrate surface during spin coating become deformed to considerable shear forces, resulting from the difference in adhesive and centrifugal forces.. Residual surfactants on the NP surface hamper solid state recrystallization between single nanoparticles. On the other hand, surfactant free NPs can undergo reorganization and eventually recrystallization at their grain boundaries. Overall, we highlight that cs-NP films suffer

significantly less disturbance from the surfactant and exhibit significantly higher crystallinity than films from SDS-NP.”

10. Finally, given that the paper is predicated on the need for large scale printing of OPVs it would be good for the authors to discuss briefly how the use of these poloxamer materials might be implemented for an actual printing process and whether these materials are already used at scale. Is this approach viable for large scale production?

Reply: We thank the referee for the insightful comments. We have not used these materials for large-scale printing yet. We hope that this technique can be adaptive to large-scale roll-to-roll or the daily life inkjet printing techniques for fabrication of OPV devices. We added the following discussion to the “**Conclusions and outlook**” section: “The demonstration of cs-NP synthesis offers a smart strategy towards industrial mass production such as roll-to-roll fabrication of OPV devices with high-efficiency and stability from eco-friendly aqueous solvent system. Most importantly, this technique enables the fabrication of high performance solar cells utilizing a desktop inkjet printer at home, which could significantly facilitate the OPV application in daily-life scenarios.”

List of revision in the main text

1. Page 5. Fig. 2e revised

e, Tyndall effect observed from P3HT;IDTBR cs-NPs dispersion and size distribution of NPs before and after storage by DLS. The dispersion sample was kept in the freezer at approximately 0 °C for 3 months.

2. Page 6 added “The constancy of averaged size of cs-NPs (**Fig. 2e**) demonstrates that the cs-NPs are highly stable and no occurrence of Ostwald ripening⁴⁶ and only minor sedimentation was observed after three months storage at ~0 °C.”
3. Page 6 added discussion on Fig. S8 “Typically, the water processing of NP does not improve the film formation. Thus, thermal annealing was applied to overcome the inhomogeneity of as-cast NP layers. **Fig. S8** exhibits that the amount of residual F127 significantly affects the microstructure formation of the deposited thin films.”

Figure S8. Optical microscope images of as cast P3HT:IDTBR films processed from cs-NP dispersion (a) without, after (b) 1 time, (c) 2 times, (d) 3 times (e) 4 times and (f) 5 times centrifugal washes.

4. Page 7 added discussion on Fig. S10, S11 and S12 “Meanwhile, the relatively weakened diffraction peaks from out-of-plane cuts has also been observed in F127-stabilized NP film with incomplete washing process (Fig. S10). NP dispersions do not show preferential orientation or anisotropic differences in crystallinity as evidenced by transmission wide-angle X-ray scattering (WAXS) and transmission small-angle X-ray scattering (SAXS) (described in Supplementary information, section “Transmission X-ray scattering”). However, AFM measurements evidence deformation of nanoparticles during film formation (Fig S12) (described in Supplementary information, section “NP deformation”), suggesting geometrical rearrangement of the single particles during spin-coating or drying.”

Figure S10. GIWAXS profiles of as cast P3HT:IDTBR films from cs-NPs after 2 times and 5 times centrifugal washes, respectively, collected from **a**, out-of-plane cuts and **b**, in-plane cuts. As shown in Fig. S8, the NP films with less than once washing step are not homogenous, which are not appropriate for GIWAXS measurement. The film with 2 times washes contains 4% of F127. The relatively low (100), (200) and (300) peaks from out-of-plane cuts indicates that the residual surfactant would deteriorate the crystallinity of NP film.

Figure S11. Transmission WAXS and SAXS profiles of SDS stabilized P3HT:IDTBR NP dispersion as well as

cs-NP dispersion after 2 and 5 times centrifugal washes.

Figure S12. **a**, AFM height image of isolated P3HT:IDTBR NPs spin-coated on silicon substrate and **b**, the extracted height profiles of the numbered 9 particles.”

5. Page 8 added Fig. S14 and the discussion “Summarizing the above observations, we suggest the following model to rationalize the shape of NPs after film deposition in **Fig. S14**. NP touching the substrate surface during spin coating become deformed to considerable shear forces, resulting from the difference in adhesive and centrifugal forces. Residual surfactants on the NP surface hamper solid state recrystallization between single nanoparticles. On the other hand, surfactant free NPs can undergo reorganization and eventually recrystallization at their grain boundaries. Overall, we highlight that cs-NP films suffer significantly less disturbance from the surfactant and exhibit significantly higher crystallinity than films from SDS-NP.”

Figure S14. Schematic overview of the structure and crystallinity of P3HT:IDTBR NPs with and without surfactant during film deposition. The NPs are smeared out along the surface and deformed into a film during spin-coating. The residual surfactant on the NP surface hampers the recrystallization of polymer and NFA during NP deformation process. For the NP after surfactant-stripping, NPs prefer to merge with each other and lead to a film with high crystallinity.

6. Page 9 added discussion on Fig. S15 “Furthermore, the PCE of cs-NP device was found to be thickness independent within an active layer between 60 and 300 nm (Fig. S15).”

Figure S15. Thickness dependence of P3HT:IDTBR cs-NP solar cells. **a**, J - V characteristics and **b**, PCE as a function of active layer thickness of cs-NP devices.

7. Page 9 added discussion of charge generation “The maximum charge carrier generation rate G_{\max} (Table 2) of those three systems indicates that the charge generation does not dominate the losses in those two control devices. The saturation current density is defined by $J_{\text{sat}} = qG_{\max}L$, where q is the elementary charge and L is the active layer thickness (Table 1).”
8. Page 9 revised the discussion of charge collection efficiency “Another method was introduced to understand the loss mechanism in THF and water (SDS) processed devices.

The exciton harvesting efficiency (η_{eh}) and charge collection efficiency (η_{cc}) were calculated based on the results of PL and IQE results (described in the Supplementary information, section “**Charge collection efficiency calculation**”). The cs-NP device exhibits an estimation of both slightly higher η_{eh} and η_{cc} as compared to both control devices, indicating a relatively efficient exciton splitting and charge collection in cs-NP system.”

9. Page 10 discussion on Fig. S22 “Light intensity-dependent V_{OC} measurements of cs-NP device suggest slightly reduced first-order recombination compared to the control devices”

Figure S22. a, V_{OC} and b, J_{SC} of the three P3HT:IDTBR solar cells as a function of light intensity.

10. Page 10 added the discussion on “burn-in loss” “Similar to the P3HT:IDTBR-based devices processed from halogenated solvents, no photoinduced “burn-in” loss was observed in water-processed solar cells, indicating the absence of film disorder under illumination.”
11. Page 10 added the discussion of low FF “Although the FFs are a little inferior than those of the optimized P3HT system, further morphology modifications on NP size, film thickness and annealing temperature would overcome the charge collection problems in those low-bandgap systems.”
12. Page 11 added the discussion on Fig. S28 “The stability data of the three systems under continuous one sun illumination is shown in **Fig. S28**. Both IDTBR-based solar cells presented PCE losses less than 5% after 1000 hours illumination, while a “burn-in” degradation (~15%) was observed for PBQ-QF:ITIC-based cs-NP solar cells.”

Figure S28. Normalized PCE evolution of **a**, PCE10:IDTBR, **c**, PBQ-QF:IDTBR and **d**, PBQ-QF:ITIC cs-NP solar cells measured under continuous one sun illumination in N_2 .

13. Page 11 added “An impressive device performance and stability was achieved via surfactant-stripping for these systems. However, we noticed that this technique might be incompatible with some material systems, for instance materials with very low solubility in THF and materials suffered from strong aggregations during NP synthesis. Therefore, the choice of materials with high solubility in THF is one essential consideration for a successful cs-NP synthesis and device fabrication.”

14. Page 12 revised Fig. 5h

Figure 5. **h**, Schematic diagram of NP (SDS) and cs-NP structures as well as morphology and charge transport in their corresponding NP devices.

15. Page 13 added “The demonstration of cs-NP synthesis offers a smart strategy towards industrial mass production such as roll-to-roll of OPV devices with high-efficiency and stability from eco-friendly aqueous solvent system. Most importantly, this technique enables the fabrication of high performance solar cells utilizing a desktop inkjet printer at home, which could significantly facilitate the OPV application in daily-life scenarios.”

16. Page 15 added the experimental information of WAXS and SAXS “WAXS and SAXS experiments were performed at the same instrument as the GIWXS/GISAXS. The suspensions were in a glass capillary. The same capillary was used for the 3 suspensions. After SAXS/WAXS measurement the same capillary was measured with water. The measurement with water was subtracted. For absolute calibration a glassy carbon standard was used. The sample to detector distance was 1597.5 mm for SAXS and 178.285 mm for WAXS.”

17. Page 15 revised the device fabrication “After drying, the substrates were coated with 40 nm of ZnO (Nanograde, N-10) by doctor-blading and then annealed at 85 °C for 5 min. Aqueous ink with 80 mg/mL P3HT/o-IDTBR, 50 mg/mL PCE10/o-IDTBR, PBQ-QF/o-IDTBR or PBQ-QF/ITIC particle ink was then spin-coated at 1000 rpm onto ZnO surface under ambient atmosphere.”

18. Page 17 added sections of author contributions, competing interests and data availability
“Author Contributions

C.X. and N.L. conceived and developed the idea. C.X. designed and coordinated the experiments, performed NP synthesis and characterization, performed device fabrication and characterization and data analysis. T.H. and A.C. performed Photo-CELIV, TPV & CE and light intensity dependency measurement and analysed the data. T.H. performed stability test of devices. X.T. performed SEM measurements and analysed the data. W.G., I.S. and T.U. performed GIWAXS, WAXS and SAXS measurements and analysed the data. M.B. and I.M. provide the materials for experiment. A.S. and A.F.S. performed NEXAFS and data analysis. N.L. and C.J.B. supervised the project. C.X., N.L. and C.J.B. wrote the manuscript with input from all co-authors.

Competing interests

The authors declare no competing interests.

Data availability.

All data generated or analysed during this study are included in the published article (and its Supplementary Information).”

19. Page 18 moved the data of IQE, exciton harvesting efficiency and charge collection efficiency to Supplementary Information as Table S4.

REVIEWERS' COMMENTS:

Reviewer #1 (Remarks to the Author):

This manuscript is discussing very interesting research, which concerns an effective strategy to construct efficient organic photovoltaics towards practical applications. The authors have carefully answered all questions, and they included new data to support their claims. Thus, this manuscript is ready for publication.

Reviewer #2 (Remarks to the Author):

In the revised manuscript all of my comments have been addressed in a satisfactory way. So I fully recommend publication in Nature Commun

Reviewer #3 (Remarks to the Author):

The revisions that the authors have provided do not address the main concerns of the referee nor do they in several instances make sense grammatically.

The authors response to the referees document has been edited to include further recommended revisions.

Reviewer #3:

This paper reports the application of poloxamers as surfactants, with temperature dependent micelle formation, in the fabrication of nanoparticulate organic photovoltaic (NP-OPV) devices. Using these materials in conjunction with highly performing donor-acceptor blends, the authors report the highest efficiencies for NP-OPV devices to date. The authors' conclusions are well-supported by extensive experimental data from a wide range of experimental techniques. The paper makes a new and substantial contribution to the field of NP-OPV research and warrants publication in Nature Communications once the following recommended revisions are addressed.

1. Page 3 L64-66: The last sentence in the paragraph referring to inkjet printing of OPV at home is not really warranted at this stage and should be removed.

Reply: We agree with the reviewer that this sentence might be inappropriate in this paragraph. The following sentence was removed in the revised manuscript.

“Most importantly, the development of a water-based OPV ink enables the fabrication of customized devices utilizing a desktop inkjet printer at home, which could significantly facilitate the application of OPV in daily-life scenarios.”

2. Page 8 L234: The word “amazingly” should be removed.

Reply: The word was removed in the revised manuscript.

3. Page 9 L237-239: The authors state that the centrifugal washing played a critical role in the NP-OPV fabrication. They need to elaborate on this statement and explain how they understand the how the washing process determines device performance.

Reply: We added the following sentence in section “**Characterization of cs-NP-based organic solar cells**” to explain how the washing step affects the device performance.

“As shown in **Fig. S16** the devices processed by dispersion with less washing steps remain more residual surfactant, resulting in losses in FFs and PCEs.”

The added sentence does not make sense grammatically. In addition, the authors need to specify how the presence of residual surfactant is affecting the FF and PCE in reference to the current literature on this topic.

4. Page 9 L261: Correct “charrier” to “carrier”

Reply: This word was corrected.

5. Page 9 L267: Replace “exhibits” with “shows”.

Reply: This word was corrected.

6. Page 9/10: The discussion around recombination dynamics is confusing. The authors argue that the recombination order ($R > 2$) reflects increased trapping and thus $R_{SDS} > R_{cs-NP}$ is consistent with residual SDS creating additional trap states. However, they do not discuss explicitly the fact that $R_{BHJ} > R_{SDS} > R_{cs-NP}$ implying that the greatest trap density occurs for the BHJ system and yet the performance of the BHJ system is consistently intermediate between the SDS and cs-NP systems as measured by all of the characterisation techniques. The authors need to discuss and clarify.

Reply: We agree that the BHJ device indeed has the highest value of R , which indicates the strong trapping effects in THF-processed P3HT:IDTBR films. The fast decay of charge carriers as a function of charge density would cause a loss in V_{oc} . That is reason that the averaged V_{oc} value in the THF-processed film is slightly lower than the other two systems (Table 1). However, the trapping effect is not the only loss mechanism in this system. As we can see from the photo-CELIV result in Fig. 4f, the charge carrier mobility of BHJ device is intermediate between SDS and cs-NP systems, leading to a moderate J_{sc} of 8.38 mA cm^{-2} and a FF of $\sim 48\%$. Although the R value of the SDS system is lower than that of the BHJ system, the ultimate device performance is also determines by other factors, including charge carrier mobility.

This discussion needs to be corrected grammatically and then included in the text of the paper to explain the trends in the observed behaviour.

7. Page 10 L300-301: The degradation data shows that the NP-OPV devices degrade without a characteristic “burn-in” phase. Can the authors explicitly comment on this observation and its implication for the degradation mechanisms in these devices?

Reply: “Burn-in” degradation typically depends on the nature of material systems, in particular on their microstructure morphology and the miscibility between donor and acceptor. According to literature, the P3HT:IDTBR does not show “burn-in” degradation [Gasparini, N. *et al.* Burn-in Free Nonfullerene-Based Organic Solar Cells. *Adv. Energy Mater.* **7**, 1700770 (2017).], which can be attributed to the highly crystalline nature of BHJ morphology after thermal annealing as well as the good miscibility between P3HT and IDTBR. To clarify, the following sentence was added to this paragraph:

“Similar to the P3HT:IDTBR solar cells processed from halogenated solvents, no photo-induced

“burn-in” losses were observed in water-processed NP solar cells,

8. Page 10 L292 – 295: The authors state that the performance enhancement in cs-NP-OPV devices can be mainly attributed to decreased energetic disorder caused by residual surfactant. Indeed, the main conclusion of the paper is the removal of excess surfactant is the dominant improvement mechanism, through the elimination of trap sites. However, what is not clear from the paper is whether this effect could also be due to an improvement in film morphology driven by the reduced surfactant concentration. In other words, while it is clear that removing excess surfactant improves performance, is that improvement due to: (a) residual surfactant not impeding charge transport or, (b) the reduced surfactant concentration driving a new morphological or structural state of the film, or (c) a combination of (a) and (b). It is clear from the paper that the authors believe that (a) is the dominant effect (given the evidence that they have), however, what is lacking from the paper is a really detailed structural study that might identify if there are substantial changes in film structure. In the absence of this study, the authors either need to provide further justification/evidence for their conclusion that (a) is dominant or to alter their conclusions to note that (b) or (c) are also possibilities.

Reply: We fully agree with the referee that we need to clarify this issue. As shown in the figure below, the optical microscopy clearly shows that residual F127 significantly perturbs film formation. In addition, we have performed GIWAXS measurements on cs-NP film processed from NP dispersions after 2 times centrifugal washing (more F127 remained in NP films). As the cs-NPs after 2 times washing only contain 4% of F127, the (100), (200) and (300) peaks are slightly lower than the surfactant-stripped NP film from out-of-plane cuts. Based on this result, we added the following sentence to the manuscript in the section “**Water-processing of cs-NP film**”:

“**Fig. S8** exhibits that the amount of residual F127 significantly affects microstructure formation of the deposited thin films.” and “Meanwhile, the relatively weakened diffraction peaks from out-of-plane cuts has also been observed in F127-stabilized NP film with incomplete washing process (**Fig. S10**).”.

Figure S8. Optical microscope images of as cast P3HT:IDTBR films processed from cs-NP dispersion (a) without, after (b) 1 time, (c) 2 times, (d) 3 times (e) 4 times and (f) 5 times centrifugal washes.

Figure S10. GIWAXS profiles of as cast P3HT:IDTBR films from cs-NPs after 2 times and 5 times centrifugal washes, respectively, collected from **a**, out-of-plane cuts and **b**, in-plane cuts. As shown in Fig. S8, the NP films with less than once washing step are not homogenous, which are not appropriate for GIWAXS measurement. The film with 2 times washes contains 4% of F127. The relatively low (100), (200) and (300) peaks from out-of-plane cuts indicates that the residual surfactant would deteriorate the crystallinity of NP film.

Based on those measurement, it is clear that the residual surfactant would induce a distinctly different film structure. In this case, we conclude that the residual surfactant would (a) deteriorate the charge transport and also (b) cause lower crystalline films. Both factors affect the performance of the NP device.

According to the referee's comments, the sentence in L292-293 was revised to "To summarise, the significant enhancement of FF and J_{sc} in cs-NP devices is mainly attributed to the increased

charge carrier mobility and the decreased microstructural disorder caused by residual stabilizers and incomplete crystalline”.

Again, the amended sentence does not make sense grammatically. “...incomplete crystalline” what?

9. Page 10 L322-324: The authors state that the residual surfactant in non-annealed SDS-NP films deteriorates crystallinity. While it is true that there is residual surfactant and that the crystallinity is reduced it is not clear that there is evidence for a causal link between the two observations. What would be the mechanism? The authors need to provide further justification for this statement.

Reply: As discussed in the previous statement, the GIWAXS in Fig. S10 and the optical microscope images in Fig. S8 exhibit that residual surfactant F127 deteriorates crystallinity. In addition, we introduced a model for elaborating the mechanism. As shown in the figure below:

Figure S14. Schematic overview of the structure and crystallinity of P3HT:IDTBR NPs with and without surfactant during film deposition. The NPs are smeared out along the surface and deformed into a film during spin-coating. The residual surfactant on the NP surface hampers the recrystallization of polymer and NFA during NP deformation process. For the NP after surfactant-stripping, NPs prefer to merge with each other and lead to a film with high crystallinity.

The elaboration has been added to section “**Water-processing of cs-NP film**”: “Summarizing the above observations, we suggest the following model to rationalize the shape of NPs after film deposition in Fig. S14. NP touching the substrate surface during spin coating become deformed to considerable shear forces, resulting from the difference in adhesive and centrifugal forces.. Residual surfactants on the NP surface hamper solid state recrystallization between single nanoparticles. On the other hand, surfactant free NPs can undergo reorganization and eventually recrystallization at their grain boundaries. Overall, we highlight that cs-NP films suffer

significantly less disturbance from the surfactant and exhibit significantly higher crystallinity than films from SDS-NP.”

10. Finally, given that the paper is predicated on the need for large scale printing of OPVs it would be good for the authors to discuss briefly how the use of these poloxamer materials might be implemented for an actual printing process and whether these materials are already used at scale. Is this approach viable for large scale production?

Reply: We thank the referee for the insightful comments. We have not used these materials for large-scale printing yet. We hope that this technique can be adaptive to large-scale roll-to-roll or the daily life inkjet printing techniques for fabrication of OPV devices. We added the following discussion to the “**Conclusions and outlook**” section: “The demonstration of cs-NP synthesis offers a smart strategy towards industrial mass production such as roll-to-roll fabrication of OPV devices with high-efficiency and stability from eco-friendly aqueous solvent system. Most importantly, this technique enables the fabrication of high performance solar cells utilizing a desktop inkjet printer at home, which could significantly facilitate the OPV application in daily-life scenarios.”

The authors have not really addressed the issue. How could these materials be applied at scale? What is their cost (both now and potential for the future) and is this viable for large scale production? Are the low temperature conditions required feasible for large scale manufacturing – how would the authors justify their comments.

The last sentence in the paragraph referring to inkjet printing of OPV at home was not warranted in the introduction and is still not warranted in the conclusions and outlook. It should be removed.

Reviewer #3:

This paper reports the application of poloxamers as surfactants, with temperature dependent micelle formation, in the fabrication of nanoparticulate organic photovoltaic (NP-OPV) devices. Using these materials in conjunction with highly performing donor-acceptor blends, the authors report the highest efficiencies for NP-OPV devices to date. The authors' conclusions are well-supported by extensive experimental data from a wide range of experimental techniques. The paper makes a new and substantial contribution to the field of NP-OPV research and warrants publication in Nature Communications once the following recommended revisions are addressed.

1. Page 3 L64-66: The last sentence in the paragraph referring to inkjet printing of OPV at home is not really warranted at this stage and should be removed.

Reply: We agree with the reviewer that this sentence might be inappropriate in this paragraph. The following sentence was removed in the revised manuscript.

“Most importantly, the development of a water-based OPV ink enables the fabrication of customized devices utilizing a desktop inkjet printer at home, which could significantly facilitate the application of OPV in daily-life scenarios.”

2. Page 8 L234: The word “amazingly” should be removed.

Reply: The word was removed in the revised manuscript.

3. Page 9 L237-239: The authors state that the centrifugal washing played a critical role in the NP-OPV fabrication. They need to elaborate on this statement and explain how they understand the how the washing process determines device performance.

Reply: We added the following sentence in section “**Characterization of cs-NP-based organic solar cells**” to explain how the washing step affects the device performance.

“As shown in **Fig. S16** the devices processed by dispersion with less washing steps remain more residual surfactant, resulting in losses in FFs and PCEs.”

The added sentence does not make sense grammatically. In addition, the authors need to specify how the presence of residual surfactant is affecting the FF and PCE in reference to the current literature on this topic.

4. Page 9 L261: Correct “charrier” to “carrier”

Reply: This word was corrected.

5. Page 9 L267: Replace “exhibits” with “shows”.

Reply: This word was corrected.

6. Page 9/10: The discussion around recombination dynamics is confusing. The authors argue that the recombination order ($R > 2$) reflects increased trapping and thus $R_{SDS} > R_{cs-NP}$ is consistent with residual SDS creating additional trap states. However, they do not discuss explicitly the fact that $R_{BHJ} > R_{SDS} > R_{cs-NP}$ implying that the greatest trap density occurs for the BHJ system and yet the performance of the BHJ system is consistently intermediate between the SDS and cs-NP systems as measured by all of the characterisation techniques. The authors need to discuss and clarify.

Reply: We agree that the BHJ device indeed has the highest value of R , which indicates the strong trapping effects in THF-processed P3HT:IDTBR films. The fast decay of charge carriers as a function of charge density would cause a loss in V_{oc} . That is reason that the averaged V_{oc} value in the THF-processed film is slightly lower than the other two systems (Table 1). However, the trapping effect is not the only loss mechanism in this system. As we can see from the photo-CELIV result in Fig. 4f, the charge carrier mobility of BHJ device is intermediate between SDS and cs-NP systems, leading to a moderate J_{sc} of 8.38 mA cm^{-2} and a FF of $\sim 48\%$. Although the R value of the SDS system is lower than that of the BHJ system, the ultimate device performance is also determines by other factors, including charge carrier mobility.

This discussion needs to be corrected grammatically and then included in the text of the paper to explain the trends in the observed behaviour.

7. Page 10 L300-301: The degradation data shows that the NP-OPV devices degrade without a characteristic “burn-in” phase. Can the authors explicitly comment on this observation and its implication for the degradation mechanisms in these devices?

Reply: “Burn-in” degradation typically depends on the nature of material systems, in particular on their microstructure morphology and the miscibility between donor and acceptor. According to literature, the P3HT:IDTBR does not show “burn-in” degradation [Gasparini, N. *et al.* Burn-in Free Nonfullerene-Based Organic Solar Cells. *Adv. Energy Mater.* **7**, 1700770 (2017).], which can be attributed to the highly crystalline nature of BHJ morphology after thermal annealing as well as the good miscibility between P3HT and IDTBR. To clarify, the following sentence was added to this paragraph:

“Similar to the P3HT:IDTBR solar cells processed from halogenated solvents, no photo-induced

“burn-in” losses were observed in water-processed NP solar cells,

8. Page 10 L292 – 295: The authors state that the performance enhancement in cs-NP-OPV devices can be mainly attributed to decreased energetic disorder caused by residual surfactant. Indeed, the main conclusion of the paper is the removal of excess surfactant is the dominant improvement mechanism, through the elimination of trap sites. However, what is not clear from the paper is whether this effect could also be due to an improvement in film morphology driven by the reduced surfactant concentration. In other words, while it is clear that removing excess surfactant improves performance, is that improvement due to: (a) residual surfactant not impeding charge transport or, (b) the reduced surfactant concentration driving a new morphological or structural state of the film, or (c) a combination of (a) and (b). It is clear from the paper that the authors believe that (a) is the dominant effect (given the evidence that they have), however, what is lacking from the paper is a really detailed structural study that might identify if there are substantial changes in film structure. In the absence of this study, the authors either need to provide further justification/evidence for their conclusion that (a) is dominant or to alter their conclusions to note that (b) or (c) are also possibilities.

Reply: We fully agree with the referee that we need to clarify this issue. As shown in the figure below, the optical microscopy clearly shows that residual F127 significantly perturbs film formation. In addition, we have performed GIWAXS measurements on cs-NP film processed from NP dispersions after 2 times centrifugal washing (more F127 remained in NP films). As the cs-NPs after 2 times washing only contain 4% of F127, the (100), (200) and (300) peaks are slightly lower than the surfactant-stripped NP film from out-of-plane cuts. Based on this result, we added the following sentence to the manuscript in the section “**Water-processing of cs-NP film**”:

“**Fig. S8** exhibits that the amount of residual F127 significantly affects microstructure formation of the deposited thin films.” and “Meanwhile, the relatively weakened diffraction peaks from out-of-plane cuts has also been observed in F127-stabilized NP film with incomplete washing process (**Fig. S10**).”.

Figure S8. Optical microscope images of as cast P3HT:IDTBR films processed from cs-NP dispersion (a) without, after (b) 1 time, (c) 2 times, (d) 3 times (e) 4 times and (f) 5 times centrifugal washes.

Figure S10. GIWAXS profiles of as cast P3HT:IDTBR films from cs-NPs after 2 times and 5 times centrifugal washes, respectively, collected from **a**, out-of-plane cuts and **b**, in-plane cuts. As shown in Fig. S8, the NP films with less than once washing step are not homogenous, which are not appropriate for GIWAXS measurement. The film with 2 times washes contains 4% of F127. The relatively low (100), (200) and (300) peaks from out-of-plane cuts indicates that the residual surfactant would deteriorate the crystallinity of NP film.

Based on those measurement, it is clear that the residual surfactant would induce a distinctly different film structure. In this case, we conclude that the residual surfactant would (a) deteriorate the charge transport and also (b) cause lower crystalline films. Both factors affect the performance of the NP device.

According to the referee's comments, the sentence in L292-293 was revised to "To summarise, the significant enhancement of FF and J_{sc} in cs-NP devices is mainly attributed to the increased

charge carrier mobility and the decreased microstructural disorder caused by residual stabilizers and incomplete crystalline”.

Again, the amended sentence does not make sense grammatically. “...incomplete crystalline” what? 
9. Page 10 L322-324: The authors state that the residual surfactant in non-annealed SDS-NP films deteriorates crystallinity. While it is true that there is residual surfactant and that the crystallinity is reduced it is not clear that there is evidence for a causal link between the two observations. What would be the mechanism? The authors need to provide further justification for this statement.

Reply: As discussed in the previous statement, the GIWAXS in **Fig. S10** and the optical microscope images in **Fig. S8** exhibit that residual surfactant F127 deteriorates crystallinity. In addition, we introduced a model for elaborating the mechanism. As shown in the figure below:

Figure S14. Schematic overview of the structure and crystallinity of P3HT:IDTBR NPs with and without surfactant during film deposition. The NPs are smeared out along the surface and deformed into a film during spin-coating. The residual surfactant on the NP surface hampers the recrystallization of polymer and NFA during NP deformation process. For the NP after surfactant-stripping, NPs prefer to merge with each other and lead to a film with high crystallinity.

The elaboration has been added to section “**Water-processing of cs-NP film**”: “Summarizing the above observations, we suggest the following model to rationalize the shape of NPs after film deposition in **Fig. S14**. NP touching the substrate surface during spin coating become deformed to considerable shear forces, resulting from the difference in adhesive and centrifugal forces.. Residual surfactants on the NP surface hamper solid state recrystallization between single nanoparticles. On the other hand, surfactant free NPs can undergo reorganization and eventually recrystallization at their grain boundaries. Overall, we highlight that cs-NP films suffer

significantly less disturbance from the surfactant and exhibit significantly higher crystallinity than films from SDS-NP.”

10. Finally, given that the paper is predicated on the need for large scale printing of OPVs it would be good for the authors to discuss briefly how the use of these poloxamer materials might be implemented for an actual printing process and whether these materials are already used at scale. Is this approach viable for large scale production?

Reply: We thank the referee for the insightful comments. We have not used these materials for large-scale printing yet. We hope that this technique can be adaptive to large-scale roll-to-roll or the daily life inkjet printing techniques for fabrication of OPV devices. We added the following discussion to the “**Conclusions and outlook**” section: “The demonstration of cs-NP synthesis offers a smart strategy towards industrial mass production such as roll-to-roll fabrication of OPV devices with high-efficiency and stability from eco-friendly aqueous solvent system. Most importantly, this technique enables the fabrication of high performance solar cells utilizing a desktop inkjet printer at home, which could significantly facilitate the OPV application in daily-life scenarios.”

The authors have not really addressed the issue. How could these materials be applied at scale? What is their cost (both now and potential for the future) and is this viable for large scale production? Are the low temperature conditions required feasible for large scale manufacturing – how would the authors justify their comments. 
The last sentence in the paragraph referring to inkjet printing of OPV at home was not warranted in the introduction and is still not warranted in the conclusions and outlook. It should be removed.